# Cross-domain metabolic interactions link *Methanobrevibacter smithii* to colorectal cancer microbial ecosystems

Rokhsareh Mohammadzadeh[1], Alexander Mahnert [ID][1], Tamara Zurabishvili[1], Lisa Wink[1], Christina Kumpitsch [ID][1], Hansjoerg Habisch [ID][2], Jannik Sprengel [ID][3,4], Klara Filek [ID][1], Polona Mertelj[1], Dominique Pernitsch[5], Kerstin Hingerl[5], Marija Durdevic[6,7], Gregor Gorkiewicz [ID][6,8], Christian Diener [ID][1], Alexander Loy [ID][9], Dagmar Kolb[5], Christoph Trautwein [ID][3,4,10,11,12], Tobias Madl [ID][2,8] & Christine Moissl-Eichinger [ID][1,8] ✉

The human gut is colonized by trillions of microbes that influence the health of their human host. Whereas many bacterial species have now been linked to a variety of different diseases, the involvement of Archaea, an evolutionarily distinct group of microbes, in human disease remains elusive. By analyzing 19 independent clinical studies, we demonstrate that associations between Archaea and human diseases are widespread yet highly heterogeneous, with a pronounced and consistent enrichment of *Methanobrevibacter smithii* in colorectal cancer (CRC) patients. Metabolic modelling and in vitro co-culture identified distinct mutualistic interactions of *M. smithii* with CRC-causing bacteria such as *Fusobacterium nucleatum*, including metabolic enhancement. Metabolomics further reveal archaeal-derived compounds with tumor-modulating properties. Together, our results provide mechanistic insights into how the human gut archaeome may participate in CRC-associated microbial networks through metabolic cooperation with bacteria.

Alterations in the gastrointestinal microbiome have been implicated in a wide range of diseases, including inflammatory bowel diseases (IBD), metabolic disorders like obesity and type 2 diabetes (T2D), cardiovascular conditions, and neurodegenerative or neuropsychiatric diseases, including Parkinson's disease (PD), Alzheimer's disease (AD), schizophrenia (SCZ), and multiple sclerosis (MS)[1–8]. Establishing mechanistic links between specific microbes and disease phenotypes

remains a major challenge due to the inherent complexity and interdependence of microbial ecosystems. Microbiomes are shaped by intricate networks of cross-feeding, competition and host interactions across domains of life, including a complex interplay of bacteria, fungi, viruses, and archaea.

Archaea, in particular methane-forming representatives, can constitute up to 4% of the gut microbiome[9], but remain severely under-

[1]Diagnostic and Research Institute of Hygiene, Microbiology and Environmental Medicine, Medical University of Graz, Graz, Austria. [2]Medicinal Chemistry, Otto Loewi Research Center, Medical University of Graz, Graz, Austria. [3]Core Facility Metabolomics, Medical Faculty University of Tübingen, Tübingen, Germany. [4]M3 Research Center for Malignome, Metabolome & Microbiome, Medical Faculty University of Tübingen, Tübingen, Germany. [5]Core Facility Ultrastructure Analysis, Medical University of Graz, Graz, Austria. [6]Institute of Pathology, Medical University of Graz, Graz, Austria. [7]Core Facility Computational Bioanalytics, Center for Medical Research, Medical University of Graz, Graz, Austria. [8]BioTechMed, Graz, Austria. [9]Division of Microbial Ecology, Centre for Microbiology and Environmental Systems Science, University of Vienna, Vienna, Austria. [10]Department of Preclinical Imaging and Radiopharmacy, Werner Siemens Imaging Center, University Hospital Tübingen, Tübingen, Germany. [11]Cluster of Excellence CMFI (EXC 2124) "Controlling Microbes to Fight Infections", Eberhard Karls University of Tübingen, Tübingen, Germany. [12]Cluster of Excellence iFIT (EXC 2180) "Image Guided and Functionally Instructed Tumor Therapies", University of Tübingen, Tübingen, Germany. ✉e-mail: christine.moissl-eichinger@medunigraz.at

researched. The most abundant genus is *Methanobrevibacter*, whose representatives are generally considered commensal and have not been linked to pathogenicity in humans or other hosts[10]. Yet, their metabolic activity, primarily the consumption of bacterial fermentation products ($H_2$, $CO_2$) to produce methane, positions them as key microbial interactors[11]. *Methanobrevibacter* abundance has been associated with beneficial outcomes, including higher short-chain fatty acid levels, reduced body mass index (BMI), and increased longevity[9,12–14]. Archaeal depletion has been observed in several disorders, such as IBD[15], obesity[16], and irritable bowel syndrome with diarrhea (IBS-D)[17,18]. Conversely, elevated *Methanobrevibacter* levels have also been linked to constipation-dominant IBS (IBS-C)[19], intestinal methanogen overgrowth (a small intestinal bacterial overgrowth subtype)[20,21], and in some reports, colorectal cancer (CRC)[22]. These associations suggest that methanogens may modulate host physiology both directly, through metabolite production, and indirectly, by shaping bacterial activity through otherwise inhibiting $H_2$ removal.

Despite their potential significance, archaea are largely neglected in most microbiome studies due to methodological limitations. Standard 16S rRNA gene-targeted primers lack archaeal coverage, reference genome databases remain incomplete, cultivation is tricky, and many computational pipelines are not optimized to detect archaeal signatures[23,24]. Most existing data on archaea come from using low-resolution methods, such as breath methane testing or 16S rRNA gene profiling, which cannot resolve species-level taxa or infer metabolic functions. As a result, key hypotheses remain untested: that distinct diseases may induce characteristic shifts in gut archaeome (the archaeal community residing in the human gut); that methanogens may be selectively enriched or depleted depending on the disease context; that certain archaeal taxa could serve as reliable, disease-specific biomarkers across diverse cohorts; and that archaeal metabolites, whether unique to this domain or produced via syntrophic interactions, may exert biologically relevant effects on host physiology and disease processes.

To address these knowledge gaps, we performed a multi-level study, starting with the first comprehensive meta-analysis of the human gut archaeome across multiple diseases, leveraging high-quality shotgun metagenomes from around 3000 fecal samples derived from 19 studies spanning 12 countries. This dataset covered a spectrum of conditions: CRC, T2D, CD, UC, MS, AD, SCZ, and PD. Using a unified analytical framework, we systematically quantified archaeal prevalence, identified disease-specific patterns, and assessed the significance of these associations across cohorts while adjusting for major confounders, including age, sex, and BMI, when possible. Observed correlations with CRC were further addressed by genome-scale metabolic modeling and archaeal-bacterial co-culture experiments, involving CRC-associated pathogens, such as *Fusobacterium nucleatum*. Functional interactions were resolved by metabolomics, revealing that *M. smithii* engages in complex exchanges that have the potential to shape the tumor microenvironment.

Our findings position *M. smithii* as a potentially active metabolic and ecological contributor within CRC-associated microbial networks. By redefining its role from a passive $H_2$ scavenger to a dynamic modulator of microbial interactions and host-relevant metabolites, this study highlights the importance of integrating archaeal functions into microbiome-disease frameworks.

## Results

### Systematic reprocessing of human gut metagenomes enables standardized archaeal profiling across diverse studies

We systematically collected, reprocessed, and reanalyzed raw microbiome datasets, selecting only those studies that provided publicly accessible metagenome sequencing data (in FASTQ or FASTA format) derived from stool samples, accompanied by disease metadata (i.e., case versus control classifications) for at least 20 subjects per

category. Out of an initial pool of 627 studies, 573 were excluded after abstract screening, resulting in 54 eligible studies. Subsequent refinements led to the exclusion of an additional 35 studies due to missing metadata, inconsistencies between sample identifiers in metadata and sequencing files, unavailable data or full texts, or the use of 16S rRNA amplicon sequencing instead of metagenomic shotgun sequencing, the latter being essential for our meta-analysis due to its superior species-level resolution for archaea. Ultimately, 19 studies were retained for meta-analysis (Fig. 1a and Supplementary Fig. 1).

The dataset initially comprised one fecal metagenome sample each for 3243 subjects. For the study by Zhou et al.[25], only individuals who had not received any medication for MS treatment were included in the analysis to minimize confounding effects[26]. Additionally, for datasets where antibiotic use was explicitly reported in the metadata (e.g., Franzosa et al. study[27] and Wallen et al. study[28]), we excluded those samples to avoid potential microbiome alterations due to antibiotic exposure. To minimize confounding effects of sex, age, and BMI, factors known to influence both the archaeome and microbiome[28–31], we matched case-control samples separately within each study whenever possible. In the studies conducted by Yu et al. and Jo et al.[32,33], case-control sample matching based on these covariates was not feasible due to incomplete metadata. However, both studies reported no significant differences in sex, age, or BMI between case and control groups. In contrast, case-control samples in datasets from Feng et al., Boktor et al., and Bedarf et al.[34–36] had already been pre-matched for sex, age, and BMI. In Franzosa et al. study[27], case-control matching was performed solely by age, as BMI and sex information were not available. For all remaining 13 studies, we applied case-control matching based on sex, age (±5 years), and BMI (±3 units), resulting in a final dataset of 2214 samples (without combining the datasets) (Supplementary Data 1 and Fig. 1b).

A detailed summary of the included studies, covering study design, geographic origin, sample sizes, processing methods, and sequencing protocols, is provided in Supplementary Data 1. While the studies utilized different DNA extraction kits, sequencing platforms, and sample accession numbers, each study maintained methodological consistency within its own framework. To ensure consistency, metagenomic data from all included studies were uniformly processed following a standardized protocol, and each study underwent independent analysis to avoid the batch effect (Fig. 1c). To gain additional insight into diseases represented by multiple cohorts, datasets were also analyzed in pooled form. Post-correction PERMANOVA (Bray−Curtis dissimilarity) analyses confirmed a substantial reduction in inter-study variance across all disease cohorts after batch correction: from $R^2 = 0.154$ to $R^2 = 0.080$ ($p < 0.001$) for CRC, from $R^2 = 0.055$ to $R^2 = 0.011$ ($p = 0.001$) for pre-AD, and from $R^2 = 0.144$ to $R^2 = 0.087$ ($p = 0.001$) for PD datasets. Due to the lack of consistent covariate information across studies, case-control matching based on the aforementioned metadata was not feasible in the combined dataset.

### Alpha and beta diversity of the gut archaeome is cohort- and disease-dependent

To investigate differences in archaeal composition between disease (case) and control groups, we performed a comprehensive re-analysis of multiple independent datasets. This approach enabled the identification of both shared and disease-specific archaeal taxa across different conditions.

To investigate the relationship between gut metagenome archaeal composition and disease, we assessed the beta diversity of the samples using Bray−Curtis distance to quantify microbial community dissimilarities, and principal coordinate analysis (PCoA) was employed to visualize clustering patterns based on species abundances derived from metagenomic shotgun sequencing. Associations between archaeome beta diversity and disease states were identified in CRC and pre-AD. Among CRC studies, 4 out of 9 (conducted by Feng et al.[34]

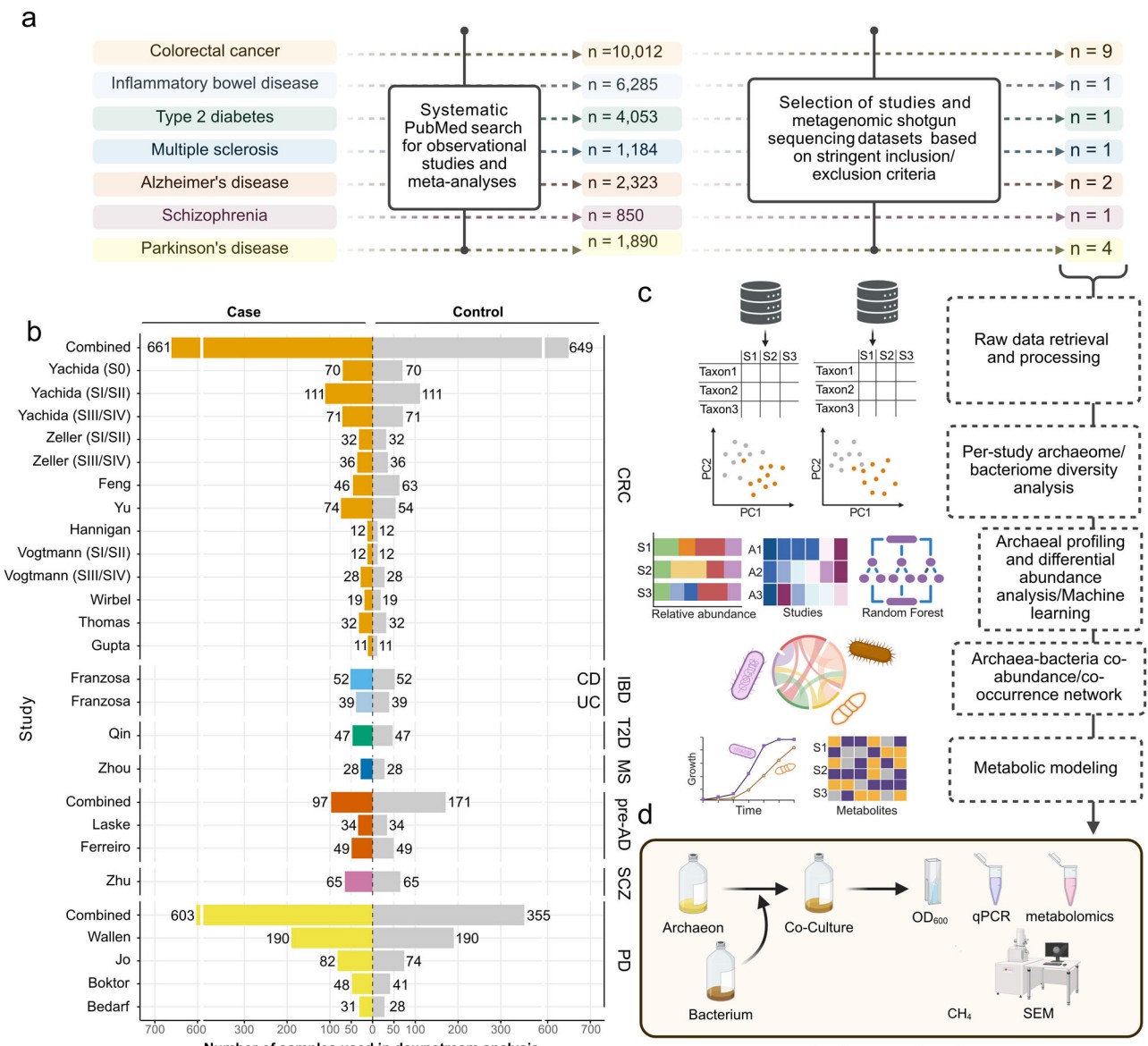

**Fig. 1 | Schematic overview of the study. a** Metagenomic sequencing datasets were selected using stringent inclusion and exclusion criteria. **b** Nine studies were retained for downstream analysis, and samples were adjusted for confounders, such as sex, age, and BMI, when possible. For diseases represented by more than one dataset (CRC, pre-AD, and PD), each dataset was first analyzed individually, and subsequently, the datasets were combined to enable integrative analyses. **c** Raw reads were retrieved, quality-controlled, and filtered to remove human sequences, followed by taxonomic classification using Kraken2 and Bracken. For pooled datasets, MMUPHin was performed for batch-effect removal. Archaeal profiles were then extracted, and differential abundance testing was performed. Machine learning was additionally performed for the pooled CRC dataset in order to assess the diagnostic potential of archaeal compositional differences in CRC. Co-

abundance correlations between *Methanobrevibacter smithii* and CRC-associated bacterial taxa were assessed, and their co-occurrence and presence–absence patterns were further examined to evaluate potential ecological associations. Metabolic modeling was used to explore metabolite exchange between *Methanobrevibacter smithii* and CRC-associated bacteria. **d** In vitro experiments were conducted to understand the dynamics between *M. smithii* and CRC-associated bacteria. CRC colorectal cancer, CD Crohn's disease, UC ulcerative colitis, T2D type 2 diabetes, MS multiple sclerosis, pre-AD pre-Alzheimer's disease, SCZ schizophrenia, PD Parkinson's disease. Figure **a**, **c**, **d** were created in BioRender (Neumann, C. (2026) https://BioRender.com/r338i6l). **b** Visualization was performed using the ggplot2 package, and axis breaks were introduced using ggbreak[113,114].

$(R^2 = 0.029, p = 1E\text{-}03)$, Yu et al.[33] $(R^2 = 0.036, p = 4E\text{-}03)$, Thomas et al.[37] $(R^2 = 0.064, p = 1.4E\text{-}02)$, and Gupta et al.[38] $(R^2 = 0.368, p = 9E\text{-}03))$ showed significant divergence. For pre-AD, both studies analyzed (Laske et al.[39], $R^2 = 0.037, p = 4E\text{-}02$; Ferreiro et al.[40], $R^2 = 0.040, p = 1E\text{-}02$) demonstrated archaeal compositional differences (Supplementary Fig. 2). No detectable archaeal community variations were found when comparing cases to controls for CD, UC, T2D, SCZ, or PD (Supplementary Fig. 2).

Interestingly, in CRC studies where archaeal beta diversity exhibited significant case-control differences, bacterial beta diversity followed

a similar trend (Supplementary Figs. 2 and 3), suggesting a potential relationship between archaeal and bacterial communities in CRC within the gut, while this trend was not observed in pre-AD. Moreover, in other datasets, including two CRC studies (Zeller et al.[41]; both stages I/II and III/ IV; and Wirbel et al.[42]), IBD (both UC and CD), three PD studies (Wallen et al.[28], Jo et al.[32], and Boktor et al.[35]), and the SCZ study, bacterial beta diversity showed distinct clustering between cases and controls, whereas archaeal beta diversity remained unchanged.

Overall, these findings suggest that archaeal beta diversity differences in disease states are not consistently observed across studies.

This variability may stem from the lower relative abundance of archaea, which limits statistical power, or from the possibility that archaea are less responsive to disease-related microbiome shifts compared to bacteria in these datasets. Furthermore, differences in cohort composition, geographic and environmental factors, patient individuality, dietary habits, different DNA extraction protocols, as well as variances in sequencing methodologies may have contributed to the inconsistencies observed across studies, potentially influencing archaeal beta diversity outcomes[24].

Analysis of alpha diversity using the Shannon index revealed significant reductions in archaeal diversity in only one of the nine CRC studies, namely the Wirbel et al. study[42], where cases had lower diversity compared to controls ($p = 3.1E-2$), while in another study (Gupta et al.[38]), CRC patients showed higher Shannon diversity compared to the control ($p = 3.3E-2$). In contrast, UC cases (Franzosa et al. study[27]) exhibited significantly reduced archaeal diversity compared to controls ($p = 4.5E-2$), suggesting a potential link UC and alterations in the archaeal community (Supplementary Fig. 4).

Notably, in studies where significant archaeal Shannon index reductions were observed, bacterial Shannon index differences followed a parallel trend, with cases showing reduced diversity compared to controls (Supplementary Figs. 4 and 5). However, this consistent relationship was not observed across all studies of the same disease, nor in diseases for which only a single dataset was available. These findings again indicate the study-dependent shifts in archaeal diversity.

## Predominant gut-associated archaea show an increasing trend in certain diseases, such as CRC

Next, we investigated if specific archaeal species are more prevalent in disease compared to the control. We first analyzed the combined dataset to identify overall trends, and then examined each study separately (to account for study-specific confounders), to assess the composition of the archaeome, and compared the presence and relative abundance of archaeal species in healthy and diseased individuals.

In total, our analysis identified eight validly described archaeal species, five taxa assigned to provisional species within known genera (e.g., Methanobrevibacter_A_sp900769095), and additional genomes with low taxonomic resolution (here referred to as unclassified), defined by the unified human gastrointestinal genome (UHGG) reference catalog.

The gut archaeome was consistently dominated by various species of the genus *Methanobrevibacter*, particularly Methanobrevibacter_A_smithii (UHGG representative of *M. smithii*), Methanobrevibacter_A_smithii_A (UHGG representative of *M. intestini*), and Methanobrevibacter_A_sp900766745. These species were universally detected across all studies, regardless of the participants' disease status. Additional *Methanobrevibacter* species, including Methanobrevibacter_A_woesei, Methanobrevibacter_A_sp900769095, and Methanobrevibacter_A_oralis, were observed at lower abundances but were consistently detected (Fig. 2a). These observations suggest that members of the genus *Methanobrevibacter* are stable and persistent colonizers of the human gut, irrespective of host disease conditions.

Aside from *Methanobrevibacter*, substantial contributions to the gut archaeome included species from the genus *Methanosphaera*. Methanosphaera_sp900322125 was detected in nearly all studies and patient groups (Fig. 2a).

Further diversity within the gut archaeome was represented by species from the genera *Methanomassilicoccus*, *Methanomethylophilus*, and *Methanobacterium*. Notably, Methanomassilicoccus_luminyensis was exclusively identified in PD patients, appearing in three out of four PD-related studies (Wallen et al.[28], Boktor et al.[35], and Bedarf et al.[36] studies) (Fig. 2a).

To explore disease-specific variations, we first evaluated the combined dataset to identify general disease-associated patterns, when possible, and then compared cases and controls within individual studies to account for study-specific effects. Differential abundance analysis (CLR + Wilcoxon, BH-FDR) of archaeal species were conducted in the context of the whole microbiome to account for their compositional relationships with the broader bacterial community. Among the five most prevalent archaeal species, Methanobrevibacter_A_smithii demonstrated significant (CLR + Wilcoxon, *p*-adjusted <0.05) differences across multiple studies (Supplementary Data 2 and 3).

A potential association of *Methanobrevibacter* was observed for neurological disorders. Methanobrevibacter_A_smithii exhibited significant enrichment in SCZ patients (*p*-adjusted = 2.10E-02). Indeed, the association of higher abundances of *Methanobrevibacter* with cognitive impairment and SCZ has been reported in patients in one study[43], whereas in healthy individuals, high methanogen phenotypes showed improved cognitive function in another study[3]. This archaeal species showed a markedly higher abundance in the combined PD datasets (*p*-adjusted = 3.64E-08). However, after subject matching, this association was no longer statistically significant within the individual datasets, although three out of four studies (Wallen et al.[28] Jo et al.[32] and Boktor et al.[35]) still exhibited an increased abundance of this species. Increased transcriptional activity of *M. smithii* has also been previously linked to alterations in microbial metabolism in the gut of PD patients[44]. The other prevalent member of the *Methanobrevibacter* genus, Methanobrevibacter_A_smithii_A (*M. intestini*), similarly displayed significant enrichment in PD patients in the combined analysis (*p*-adjusted = 9.55E-08). As with the previous species, the significance was lost after matching cases and controls in the individual studies, yet a consistent, though nonsignificant, enrichment trend was observed across all four cohorts. Similar patterns were observed for Methanomassilicoccus_luminyensis and Methanomassiliicoccus_A intestinalis, which were significantly enriched in PD patients in the pooled dataset (*p*-adjusted = 4.94E-02 and *p*-adjusted = 5.69E-04, respectively), but lost statistical significance in the case-control matched, study-specific analyses.

An opposite trend was observed in CD, where several archaeal species were significantly reduced under disease conditions. This depletion included Methanobrevibacter_A_smithii (*p*-adjusted = 2.90E-03), Methanobrevibacter_A_smithii_A (*M. intestini*) (*p*-adjusted = 1.10E-03), Methanobrevibacter_A_oralis (*p*-adjusted = 2.47E-02) and Methanobrevibacter_A_sp900769095 (*p*-adjusted = 2.47E-02), while Methanobrevibacter_A_sp900766745 (*p*-adjusted = 1.72E-02) was shown to be higher compared to healthy controls (Fig. 2b and Supplementary Fig. 6).

In the pooled CRC dataset, Methanobrevibacter_A_smithii showed a significant enrichment in CRC patients compared to controls (*p*-adjusted = 9.78E-05). Consistently, in the individual studies, this trend remained evident (except for Thomas's study[37]), reaching statistical significance in two cohorts (Feng et al.[34], *p*-adjusted = 1.56E-03; Gupta et al.[38], *p*-adjusted = 4.85E-03) (Fig. 2b). In the cohorts where CRC staging could be assessed after case-control matching (Yachida et al.[45], Zeller et al.[41], and Vogtmann et al.[46]), we observed a stage-dependent trend in the abundance of Methanobrevibacter_A_smithii. While in the Yachida et al. cohort[45], *M. smithii* was already enriched in stage 0 lesions, this archaeon tended to be depleted in early-stage CRC (stages I–II), but showed enrichment in advanced stages (III–IV). Similar correlations between CRC progression and *Methanobrevibacter* abundance in the gut has also been reported before[47,48].

Methanobrevibacter_A_smithii_A (*M. intestini*) similarly displayed significant enrichment in CRC patient groups in studies by Feng et al.[34] (*p*-adjusted = 2.61E-03) and Gupta et al.[38] (*p*-adjusted = 4.93E-03). Methanobrevibacter_A_sp900766745 was also enriched in CRC patients in Feng et al. study[34] (*p*-adjusted = 3.89E-02) (Fig. 2b).

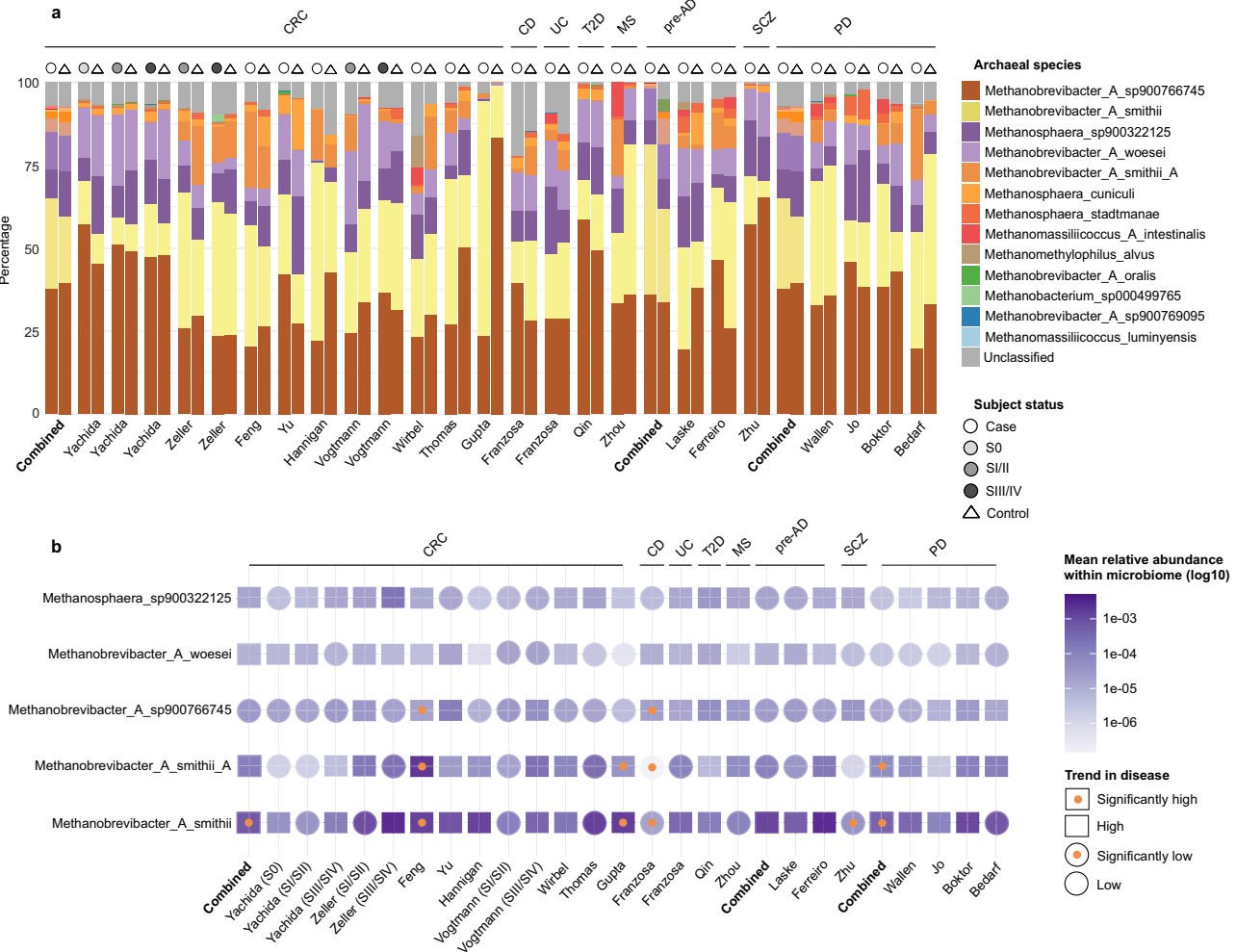

**Fig. 2 | Archaeal community profiles and differential abundance analysis across disease cohorts. a** Stacked bar plots showing the relative abundances of archaeal species across shotgun metagenomic datasets, stratified by disease and study. Within each study, samples are grouped according to disease status (cases vs. controls). For the Yachida et al.[45], Zeller et al.[41], and Vogtmann et al.[46] cohorts, where sufficient information and sample sizes were available, subjects were further stratified by colorectal cancer (CRC) stage. **b** Five abundant archaeal species are represented, each with a symbol indicating the direction of abundance change between disease and control groups: squares denote higher abundance in disease, and circles denote lower abundance in disease. An orange dot inside the symbol indicates a statistically significant difference (FDR-adjusted <0.05), while empty symbols represent nonsignificant trends. Archaeal species are represented by their corresponding UHGG (unified human gastrointestinal genome) identifiers. Differential abundance was assessed using CLR-transformed (centered log-ratio) data and two-sided Wilcoxon rank-sum test with FDR (false discovery rate) correction. CRC colorectal cancer, S0 stage 0, SI/SII stage I/II, SIII/SIV stage III/IV, CD Crohn's disease, UC ulcerative colitis, T2D type 2 diabetes, MS multiple sclerosis, pre-AD pre-Alzheimer's disease, SCZ schizophrenia, PD Parkinson's disease. Source data are provided as a Source data file.

Considering that not all studies might have used an archaea-suitable methodology[24], these associations are a strong indicator for a potential association of *Methanobrevibacter* species with CRC. Similar positive associations of *Methanobrevibacter* signatures with CRC have been previously reported in 16S rRNA gene-based studies[49], and a recent meta-analysis of shotgun metagenomic data[22].

**Metabolic cross-feeding between *M. smithii* and CRC-associated bacteria suggests functional integration**

A consistent, significant, and positive association between elevated *M. smithii* abundance and CRC was demonstrated across three independent studies (studies by Feng et al.[34], Yu et al.[33], and Gupta et al.[38]), and was also confirmed by a recent CRC meta-analysis[22] and our study (Fig. 2b). To evaluate whether observed differences in *M. smithii* abundance were detectable within a multivariate microbiome classification framework, we applied a random forest (RF) classification model trained on microbial relative abundances of the pooled dataset. Across all folds, the model achieved a mean receiver operating

characteristic (ROC) area under the curve (AUC) of 0.74 ± 0.03, indicating a stable discriminative performance between CRC and control microbiomes (Supplementary Fig. 7a). The learning curve showed convergence between training and validation AUC values, confirming generalization without overfitting.

Notably, the archaeon Methanobrevibacter_A_smithii ranked among the top discriminatory features, indicating that archaeal abundances substantially contributed to the signal captured by the model (Supplementary Fig. 7b). SHAP analysis further characterized model behavior, indicating that Methanobrevibacter_A_smithii contributed to the random forest's classification of CRC and control samples (Supplementary Fig. 7c). This observation is consistent with a recent meta-analysis, in which *M. smithii* was also identified among the top features in a CRC diagnostic model[22]. The consistent identification of *M. smithii* as both significantly enriched in CRC (Fig. 2b) and as a high-impact classifier feature (Supplementary Fig. 7b, c) suggests that this archaeon contributes to the model's discrimination between CRC and control samples.

Given that methanogenic archaea rely on metabolic by-products of bacterial fermentation, we aimed to explore the ecological and functional interactions between *M. smithii* and bacterial species previously associated with CRC.

We curated a set of twelve bacterial taxa identified as CRC microbial biomarkers based on a literature review (Supplementary Data 4). Taxa are referred to by their original names in UHGG v.2.0.1 datasets, with updated nomenclature (GTDB r226) provided in parentheses where applicable. The selected taxa include Prevotella_copri_A (updated to Segatella_copri), Mediterraneibacter_torques, Prevotella_intermedia, Peptostreptococcus_stomatis, Porphyromonas_asaccharolytica, Parviromonas_micra, Gemella_morbillorum, Clostridium_Q_symbiosum (updated to Otoolea_symbiosa), Akkermansia_muciniphila, Escherichia_coli_D (updated to Escherichia_coli), Bacteroides_fragilis, and Fusobacterium_nucleatum.

We validated the presence and enrichment of these bacterial biomarkers in our compiled CRC dataset through differential abundance analysis. In the combined dataset, 9 of the bacterial CRC biomarkers showed significant differential abundance, whereas Prevotella_A_copri, Mediterraneibacter_torques, and Porphyromonas_asaccharolytica were not significantly enriched (Supplementary Fig. 8). Across individual studies, after we controlled for confounding factors, such as age, sex, and BMI, 10 bacterial CRC biomarker species exhibited significant differential abundance across at least two independent studies (CLR-transformed data with the Wilcoxon rank-sum test, *p*-adjusted <0.05), whereas Prevotella_A_copri and Akkermansia_muciniphila each demonstrated differential abundance in only one study (Supplementary Fig. 8). Notably, all taxa, except Mediterraneibacter_torques, appeared at least once in our selected CRC datasets, and our differential abundance findings aligned mostly with the original articles, with some small differences, which could be due differences in case-control sample matching (Supplementary Fig. 8 and Supplementary Data 5).

Correlation analysis between these CRC-associated bacterial taxa and Methanobrevibacter_A_smithii revealed predominantly positive abundance-based associations across both the combined and individual datasets, with all taxa showing positive correlations with this archaeon except for Otoolea_symbiosa, which exhibited mostly negative correlations (Supplementary Fig. 9 and Supplementary Data 6). In addition, co-occurrence analysis based on presence–absence patterns showed that Methanobrevibacter_A_smithii exhibited mainly positive co-occurrence with CRC-associated bacterial taxa, with both the frequency and magnitude of these associations being higher in CRC than in controls. Although the differences in co-occurrence strength between groups were not statistically significant, the overall trend indicates a more pronounced archaeal–bacterial co-association under disease conditions (Supplementary Fig. 10), which was in accordance with previous studies where co-occurrence of *M. smithii* and these bacterial taxa were shown[22].

The *M. smithii* genome encodes an extensive array of transporters (see transporter file in gapseq output of *M. smithii* in Data availability), including those for amino acids (e.g., aspartate, arginine, glutamate, glutamine, or histidine), but also organic acids (e.g., succinate), indicating that the archaeal-bacterial metabolite exchange might go well beyond $H_2$ and $CO_2$ transfer. Experimental cultivation of *M. smithii* ALI (mono-culture in rich MS medium, Supplementary Data 7) confirmed the uptake of several amino acids (leucine, valine, isoleucine, alanine, arginine, glutamic acid, methionine, asparagine, lysine, cystine, glycine, threonine, tyrosine, histidine, phenylalanine, and tryptophane) and other compounds (butyrate, propionate, lactic acid, and acetic acid etc.) from its environment (Supplementary Fig. 11 and Supplementary Data 7).

To functionally explore archaeal-bacterial cross-feeding potential, we performed in silico co-culture metabolic modeling using PyCoMo[50]. The models included all 12 bacterial CRC biomarkers (one in each simulated co-culture), and, as a control, the non-CRC-associated *Christensenella minuta*, given its known syntrophic interaction with *M. smithii*[14].

A key finding was the universal predicted export of succinate by all bacterial partners ($n = 13$, 100%), and the predicted import of this compound by *M. smithii* (Fig. 3). This widespread pattern suggests that succinate-mediated cross-feeding could be a conserved feature of archaeal-bacterial interactions. Succinate is a common fermentation by-product during carbohydrate fermentation, primarily from fumarate respiration, serving for electron disposal[51,52]. *M. smithii*, encoding succinate-specific transporters, succinate dehydrogenase (Sdh) and succinyl-CoA synthetase (Suc), is thus well equipped to utilize this metabolite, potentially for redox balancing or as an anaplerotic substrate in its incomplete reductive TCA cycle[53,54].

The relevance of succinate extends beyond microbial metabolism. Succinate, produced by bacteria, such as *Fusobacterium nucleatum*, has been shown to serve as a critical metabolic signaling molecule implicated in promoting CRC metastasis by activating the transcription factor STAT3, which induces epithelial-to-mesenchymal transition and enhances tumor invasiveness and metastatic potential[55].

Additionally, succinate suppresses the cGAS-interferon-β signaling pathway, reducing CD8 + T cell infiltration in tumors and consequently contributing to impaired anti-tumor immunity[56,57]. This positions succinate cross-feeding as a mechanistic link between microbiome metabolism and host oncogenic processes[58].

Beyond succinate, *M. smithii* models consistently exported riboflavin in all pairwise interactions. Riboflavin is an essential precursor for the synthesis of flavin coenzymes, flavin adenine dinucleotide (FAD) and flavin mononucleotide, which play a central role in oxidation-reduction reactions, redox metabolism[59] and consequently, *M. smithii* may be beneficial to its bacterial partners. Additional predicted export-import dynamics were observed for metabolites, such as amino acids (L-asparagine, L-glutamine), taurine, ornithine, various carbohydrates, methanol, and acetaldehyde (Fig. 3).

Notably, in interactions with CRC-associated bacteria, *M. smithii* models exhibited uptake of amino acids, including L-asparagine*, L-glutamine*, L-leucine*, L-threonine*, L-valine*, L-lysine*, L-serine*, L-arginine*, L-aspartate, L-glutamate (Fig. 3). Most of these amino acids (indicated by *) have previously been linked to CRC[55,56,60–67]. These amino acids contribute to CRC progression by supporting structural protein synthesis, providing alternative energy sources, and activating oncogenic pathways, such as mTORC1 (mechanistic target of rapamycin complex 1). Their elevated levels in tumor tissues reflect increased uptake and metabolic reprogramming by cancer cells and have been shown to be further shaped by gut microbial interactions. The uptake of these amino acids by *M. smithii* suggests a potential co-feeding pattern that may not only support microbial interactions but could also influence host metabolic or signaling pathways. Notably, while amino acid and organic acid uptake is not unique to *M. smithii*, our goal was not to establish metabolic specificity but rather to delineate potential routes of archaeal–bacterial metabolite exchange.

## Co-culture with *M. smithii* promotes bacterial growth and metabolite production linked to CRC biomarkers

To investigate the interactions between *M. smithii* and CRC-associated bacteria, we performed a series of controlled co-cultivation experiments under strictly anoxic conditions (Fig. 1d and Supplementary Fig. 12). We selected *F. nucleatum*, enterotoxigenic *Bacteroides fragilis*, and *Escherichia coli* (*E.coli*) and their well-documented roles in CRC pathogenesis. These bacteria are known to contribute to tumor development through specific virulence factors, including colibactin, cytolethal distending toxin (CDT), cycle-inhibiting factor (CIF), and cytotoxic necrotizing factor (CNF) in *E. coli*; the adhesin FadA in *F. nucleatum*; and the enterotoxin Bft in *B. fragilis*[68]. Notably, the *E. coli* strain, previously referred to as strain D, was isolated in our previous

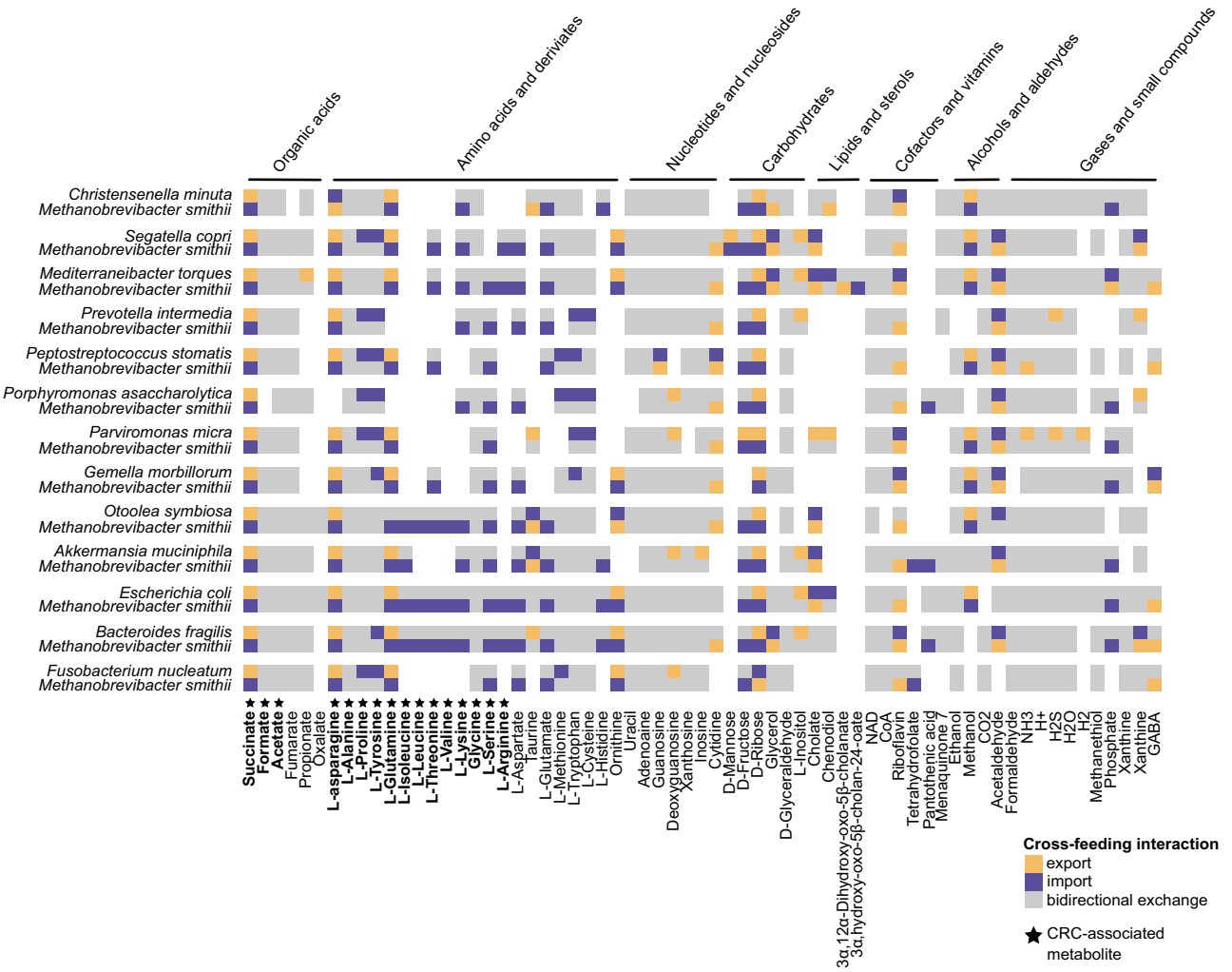

**Fig. 3 | Integration of metabolic modeling analysis in colorectal cancer (CRC) datasets.** Community-scale metabolic models show predicted cross-feeding interactions between *Methanobrevibacter smithii* and CRC-associated bacterial biomarkers in gut medium, generated using PyCoMo. Metabolite exchanges were calculated independently of growth rates for each archaeon–bacterium pair. Colors indicate metabolite export or production (yellow), import or consumption (purple), or both (gray). Only metabolites involved in cross-feeding interactions are shown. Metabolites highlighted in bold have been previously linked to CRC in the literature. Source data are provided as a Source data file.

study from a methane-producing patient, where it was found to co-occur with *M. smithii*[69]. As archaeal representative, we chose the human gut strain *M. smithii* ALI (the type strain of *M. smithii* was isolated from an anaerobic digester).

SEM analysis revealed close proximity between *M. smithii* ALI and bacterial partners in all co-culture conditions, without showing any specific morphological alterations relative to mono-cultures. All co-cultures appeared densely populated and contained numerous archaeal and bacterial cells undergoing division (Fig. 4a). Consistent with SEM observations, fluorescence microscopy revealed the presence of *M. smithii* strain ALI cells in co-cultures exhibiting the characteristic shape of this archaeal strain and $F_{420}$-based auto-fluorescence in both the mono- and co-culture conditions, indicating active growth.

Co-culture dynamics were highly specific. *M. smithii* ALI exhibited increased growth ($p = 4.1E-02$) (evidenced by elevated *mcrA* gene copy numbers) only in the presence of *B. fragilis* compared to mono-culture conditions. *F. nucleatum* ($p = 3.07E-06$) and *E. coli* ($p = 4.61E-06$) displayed significantly enhanced 16S rRNA gene copy numbers at time point 1 compared to mono-cultures, indicating archaeon-mediated stimulation of bacterial growth speed (Supplementary Fig. 13 and Supplementary Data 8). However, methane production by *M. smithii*

ALI was not significantly different between mono-cultures and none of the co-cultures (Supplementary Fig. 14). Notably, none of the co-cultures resulted in a statistically significant mutual growth benefit, suggesting asymmetrical or unidirectional metabolic support (Supplementary Fig. 13).

Metabolic profiling further elucidated the biochemical landscape of the archaeal-bacterial interactions. In agreement with in silico modeling, all co-cultures produced higher amounts of succinate compared to their mono-cultures, confirming this metabolite as a conserved cross-feeding intermediate (Figs. 3, 4b, and Supplementary Fig. 15). While metabolic alteration in *B. fragilis* and *E. coli* co-cultures with *M. smithii* ALI were relatively modest beyond succinate, the *F. nucleatum*-*M. smithii* pairing exhibited a distinct and expansive metabolic signature.

Specifically, co-cultivation with *F. nucleatum* led to a significant accumulation of succinic acid*, acetic acid, lactic acid, propionic acid, butyric acid, and a range of amino acids, including alanine*, proline*, isoleucine*, leucine*, valine* and glycine*, arginine*, and phenylalanine*. Conversely, notable consumption of methionine, glucose, choline, glycerophosphocholine, and pyroglutamic acid was observed (Supplementary Data 9, 10, and 11). Several of these amino acids (marked with *) have previously been implicated in the metabolism of CRC[55,56,60–67], reinforcing their potential functional relevance.

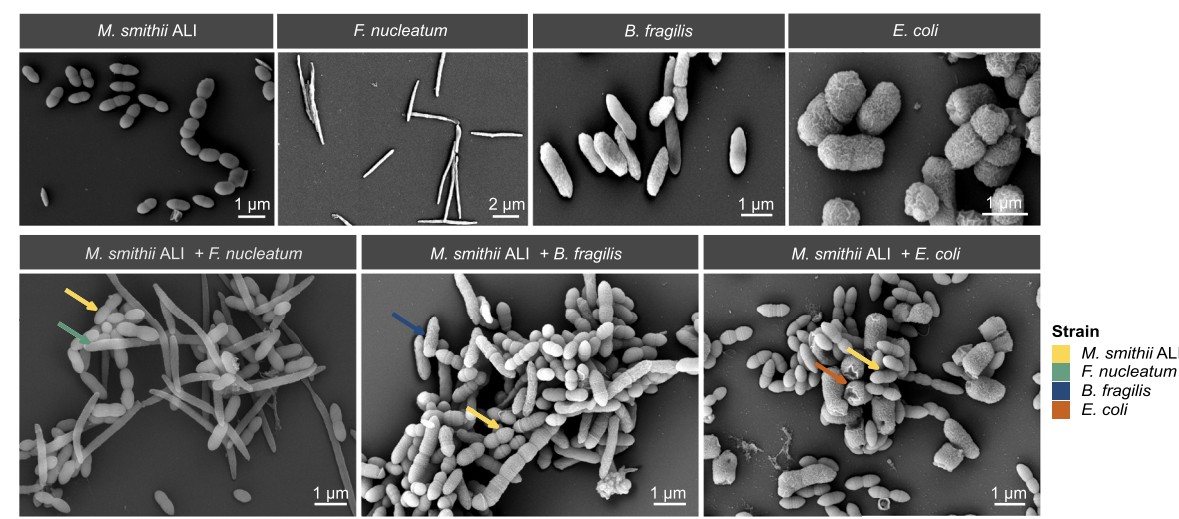

Although the co-culture results did not entirely recapitulate the in silico predictions (e.g., the predicted production of asparagine and glutamine could not be measured; Fig. 4b and Supplementary Fig. 15), both approaches converged on the identification of succinate and amino acids as key mediators of archaeal-bacterial metabolic exchange. These findings indicate *M. smithii* as an active participant in CRC-associated microbial networks, potentially shaping the tumor microenvironment through metabolite cross-feeding.

## CRC-associated compounds are prominent in the *F. nucleatum*–*M. smithii* symbio-metabolome

To further investigate the metabolites present in co-culture, we subjected the supernatant from three replicates of *M. smithii* ALI and *F. nucleatum* co-culture at time point t1 (corresponding to the highest observed growth, as shown in Supplementary Fig. 13) to detailed mass spectrometry (MS) analysis. This analysis aimed to identify metabolites specifically enriched under co-culture conditions compared to the blank control medium (MS + BHI).

**Fig. 4 | Scanning electron microscopy and NMR metabolomic profiling of *Methanobrevibacter smithii* ALI in mono- and co-culture with colorectal cancer-associated bacterial strains. a** Scanning electron microscopy (SEM) analysis of *M. smithii* ALI and the bacterial strains used in co-culture experiments, including the colorectal cancer-associated species *Fusobacterium nucleatum*, *Bacteroides fragilis*, and *Escherichia coli*. Imaging was performed for both mono-cultures and co-cultures with *M. smithii* ALI. The experiments were repeated two times independently with similar results. **b** The heatmap displays normalized area under the curve (AUC) values obtained from metabolomics analysis, representing the relative abundance of metabolites across conditions. For improved comparability and visualization, the amount of metabolites were centered, and log-ratio transformed. A metabolite was classified as enriched or depleted in co-culture only if its abundance was significantly higher or lower, respectively, compared to both *M. smithii* ALI and the corresponding bacterial mono-culture, as well as in relation

to the blank medium. For each co-culture experiment, mono-cultures of *M. smithii* ALI and the respective bacterial strain were grown in parallel under identical conditions. Both monocultures and co-cultures were performed in five biological replicates, with blank medium included in three replicates. Metabolites previously associated with colorectal cancer are highlighted in bold and indicated with an asterisk. Time points are defined as follows: t(−1): initial of the experiment prior to inoculation; t0: 24 h post-inoculation of *M. smithii* ALI; t1: 24 h post-inoculation of the bacterial strain (48 h post *M. smithii* ALI inoculation), t2: 72 h post bacterial inoculation (96 h post *M. smithii* ALI inoculation). A blank medium control was included to account for background metabolite levels. Metabolite depletion (triangle) or enrichment (circle) in co-cultures was determined using two-sided Wilcoxon rank-sum test, adjusted-*p* < 0.05 (FDR). Source data are provided as a Source data file.

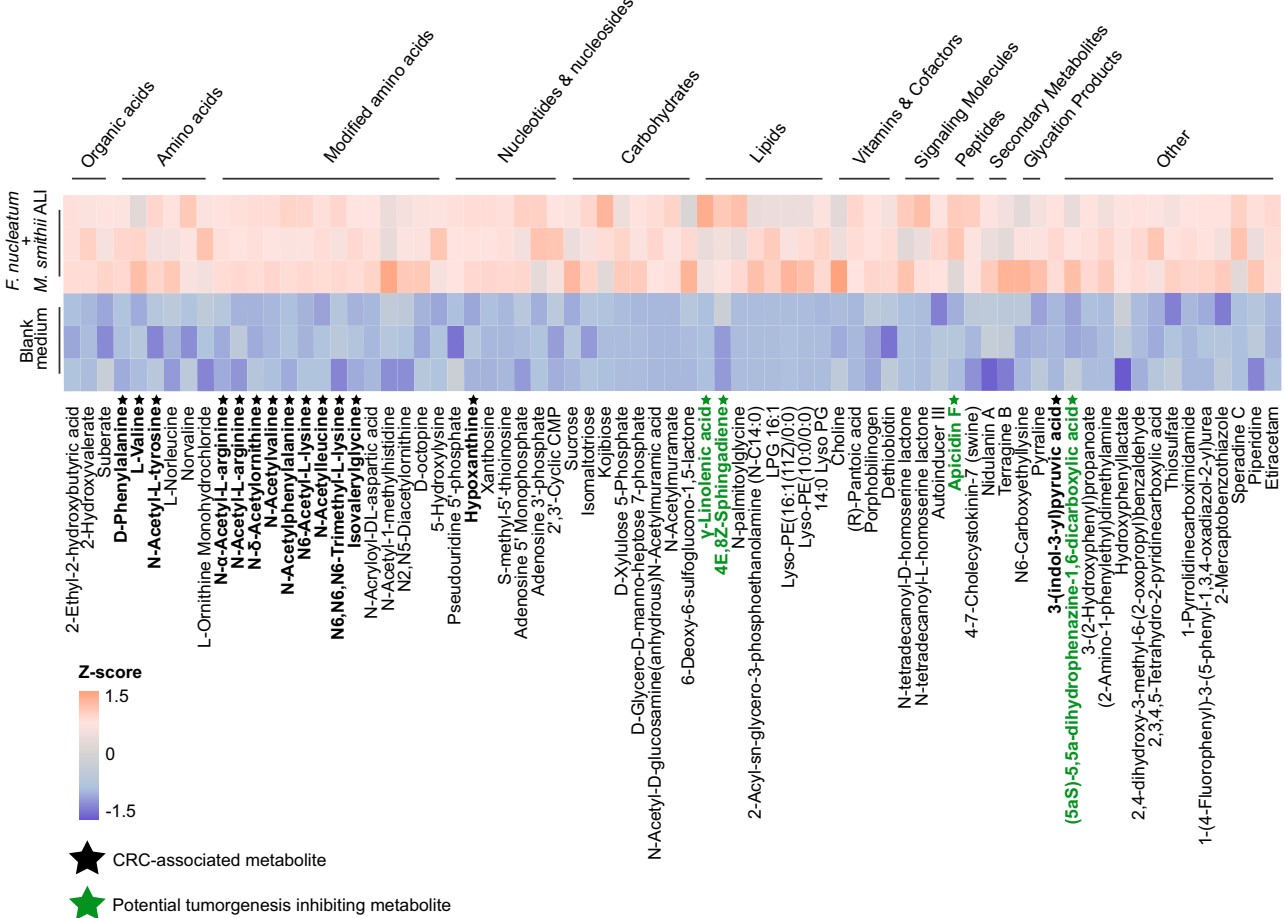

**Fig. 5 | Mass spectrometry-based metabolomic profiling of the supernatant from *Methanobrevibacter smithii* ALI + *Fusobacterium nucleatum* co-culture (triplicates).** Heatmap showing significantly increased metabolites (FC > 1.5, two-sided t-test, *p*-adjusted <0.05) detected by mass spectrometry in the co-culture of *F. nucleatum* and *M. smithii* ALI at t1 (24 h after addition of *F. nucleatum* to an established *M. smithii* ALI culture, coinciding with peak growth of both strains) compared to the blank medium (BHI + MS). Each cell represents the Z-score of the

respective metabolite, calculated across all samples as the number of standard deviations from the mean. Metabolite concentrations were normalized using PQN and log₁₀-transformed to adjust for cell count differences and dilution effects. Metabolites with reported CRC-related promoting effects are highlighted in black, while those with potential tumorigenesis-inhibiting effects are highlighted in green. Source data are provided as a Source data file.

The co-culture supernatant displayed a diverse range of metabolite classes, including organic acids, amino acids and their derivatives, nucleotides, carbohydrates, lipids, vitamins, signaling molecules, and peptides (Fig. 5 and Supplementary Data 12). Notably, several amino acids previously associated with CRC, such as phenylalanine and valine, as well as amino acid derivatives, including N-acetyl-tyrosine, N-acetyl-valine, N-acetyl-phenylalanine, N6-acetyl-L-lysine, N-acetylleucine, N6,N6,N6-trimethyl-L-lysine, and isovalerylglycine, were

significantly enriched. These compounds are derivatives of CRC-associated parent amino acids, such as tyrosine, valine, phenylalanine, lysine, leucine, and glycine, which were previously highlighted in Figs. 3, 4b, and Supplementary Fig. 11. Their presence in the MS data is consistent with findings from NMR-based metabolomics (Fig. 4b). Additionally, we detected N-α-acetyl-L-arginine, N-acetyl-arginine, and N(δ)-acetylornithine, which are metabolites involved in the arginine biosynthesis and degradation pathways in the co-culture supernatant

(Fig. 5). Arginine serves as a precursor for nitric oxide (NO) and poly-amines; while polyamines promote tumor growth[70], NO can exert either tumor-promoting or tumor-suppressive effects depending on its concentration[71].

Hypoxanthine, a key intermediate in purine metabolism, was also detected in the co-culture supernatant of *M. smithii* ALI + *F. nucleatum*. This metabolite is widely produced and utilized by both host and microbial cells. Previous studies have associated microbial hypox-anthine production, particularly by *Peptostreptococcus stomatis*, another CRC-associated bacterium (Supplementary Fig. 8 and Fig. 3), with altered gut motility and serotonin release, processes that may contribute to CRC progression[63,72–74].

Another interesting metabolite in the supernatant of the co-culture was 3(-indole-3-yl) pyruvic acid, also known as indole-3-pyruvic acid (I3P). I3P supplementation has been shown to rescue tumor cells under tryptophan starvation, identifying it as a critical oncometabolite and therapeutic target[75]. Microbial production of I3P could circumvent tryptophan restriction, suggesting tumor-microbiome metabolic cross-talk that may promote cancer progression, particularly in gut-associated cancers like CRC, where gut barrier dysfunction could be present[76,77].

Notably, several metabolites with known or potential antitumor activity were also identified in the co-culture supernatant (Fig. 5). Among these, γ-linolenic acid, previously derived from *Lactobacillus plantarum* MM89, has been shown to induce ferroptosis in tumor cells, suggesting a therapeutic mechanism for targeting CRC[78]. Another detected metabolite, 4E,8Z-sphingadiene, belongs to the sphingadiene class, has demonstrated pro-apoptotic effects in colon cancer cells and has been shown to suppress intestinal tumorigenesis in vivo, high-lighting its potential in cancer prevention[79]. Intriguingly, both γ-linolenic acid and 4E,8Z-sphingadiene were also significantly enriched in *M. smithii* ALI mono-cultures (compared to *F. nucleatum* mono-cultures), implicating the archaeon (and not the bacterial partner) as a direct source of potentially bioactive, antitumor metabolites (Sup-plementary Fig. 16 and Supplementary Data 13).

Apicidin F, a structural analog of the histone deacetylase inhibitor apicidin, was also present, which has been previously isolated from the fungus *Fusarium fujikuroi*[80]. Its detection in the supernatant of co-culture suggests the potential ability of these two strains to also pro-duce this compound (Supplementary Fig. 16). Notably, apicidin has shown anti-growth activity by inhibiting histone deacetylases in cancer cells[81]. Lastly, (5aS)-5,5a-dihydrophenazine−1,6-dicarboxylic acid, a derivative of phenazine−1,6-dicarboxylic acid, has been shown to be produced by strains, such as *Pseudomonas aeruginosa*, and has demonstrated strong anticancer activity against various cell lines, including CRC cells like HT29[82,83].

## Discussion

*Methanobrevibacter* species dominate the human gut archaeome. Their activity metabolizes $H_2$ and $CO_2$ into methane, thereby relieving lumen gas pressure, and sustains redox conditions that favor bacterial fermentation[14,69,84]. Although viewed as commensal, shifts in *Metha-nobrevibacter* abundance have been linked to various metabolic and gastrointestinal disorders, suggesting a potential role in host health[30,85]. However, most existing studies rely on low-resolution methodologies, such as breath methane testing or 16S rRNA gene sequencing, which are limited to resolve archaeal taxonomy and to link archaeal species to physiological patterns[24].

To address these limitations, we performed a meta-analysis of 2959 shotgun-metagenomic datasets, drawn from 10 studies across 12 countries, and corrected for age, sex, and BMI, when possible, for study-individual analyses. The datasets included different disorders, including CRC, T2D, (IBD) (including Crohn's disease (CD) and ulcerative colitis (UC)), MS, AD, SCZ, and PD.

Our analysis uncovered disease-specific alterations in gut archaeal communities. For instance, patients with CD showed a significant reduction in *Methanobrevibacter* spp., with the notable exception of Methanobrevibacter_A_sp900766745, which was significantly enri-ched in these patients. While the underlying mechanisms remain unclear, the reduced stool transit time in patients with CD may inter-fere with the slower growth rate of human gastrointestinal archaea.

In contrast, SCZ patients exhibited a significant depletion of Methanobrevibacter_A_smithii, which confirms previously observed correlations of methanogen abundance and neurological function[3].

In PD, we initially observed significantly higher levels of metha-nogenic taxa, including *Methanobrevibacter* species, in the pooled dataset. However, once we applied strict matching for age, sex, and BMI, these associations did not remain significant within individual studies, although most cohorts continued to show a similar direction of change. This discrepancy compared to the findings of one of the included studies (Wallen et al.[28]), who reported significant increased abundance of in PD, highlights how sensitive archaeal signals can be to study design and metadata handling. It also reinforces that robust metadata and consistent analytical approaches are essential when interpreting archaeome dynamics in disease.

Moreover, in general, variability in pipelines, reference genome databases, and cohort metadata contributes to inconsistencies across studies. These challenges highlight the need for consistent profiling frameworks and more comprehensive metadata collection, including antibiotic use and medication history, which are often overlooked but likely influence archaeal dynamics. Dietary data were also not con-sistently reported across datasets, preventing us from accounting for potential differences in eating behaviors between case and control groups, for any of the included studies.

In CRC, methanogens showed a more consistent pattern. We observed overall significant enrichment of Methanobrevibacter_-A_smithii in the pooled analysis, aligning with findings from a recent multicohort study[22]. Across individual cohorts, samples from CRC patients consistently demonstrated an increase in Methano-brevibacter_A_smithii and closely related taxa (e.g., Methano-brevibacter_A_smithii_A). While statistical significance varied between cohorts, the overall trend pointed toward the enrichment of metha-nogen in CRC. Notably, only one of the investigated studies (Gupta et al.[38]) explicitly reported increased *Methanobrevibacter* abundance in CRC, while others (e.g., Feng et al.[34]) broadly implicated archaeal overgrowth without specifying the taxonomy. In datasets with staging information, the abundance of Methanobrevibacter_A_smithii showed a nonlinear trend: elevated at stage 0, reduced in early cancer (stages I/ II), and increased again in advanced disease (III/IV). This stage-associated trajectory aligns with a recent multicohort analysis of 3,741 stool metagenomes, which identified *M. smithii* among the top species enriched in metastatic CRC and previously noted methane-producer involvement in stage IV disease[45,48]. This pattern raises the question of whether methanogen expansion precedes tumor pro-gression or reflects later shifts in the tumor-altered gut ecosystem. Notably, methanogens, including *M. smithii*, have also been associated with longevity[31], indicating that their increased abundance does not inherently imply pathogenicity. Strain-level heterogeneity among methanogens, as well as host-dependent or epigenetic interactions, may further shape their functional effects. Longitudinal and strain-resolved analyses will therefore be required.

Methane is the primary end-product of methanogen metabolism, and has been shown in animal models to slow intestinal transit[86]. Clinically, elevated baseline breath methane levels have been linked with constipation, which is a recognized risk factor for CRC[87,88]. These findings suggest a potential association between increased *M. smithii* abundance and CRC development. Notably, prior studies have repor-ted enrichment of *M. smithii* in CRC patients even after considering gut

transit time[22], implying that factors beyond motility, including host–microbe interactions or microbial cross-talk, may play a role. However, methanogen abundance patterns remain susceptible to confounders, such as constipation, which is common in CRC and can influence archaeal composition. The lack of standardized metadata, particularly regarding transit time, in existing cohorts limits our ability to fully resolve these effects and interpret associations[88].

To investigate the potential functional contributions of *M. smithii*, we further explored its interactions with CRC-associated bacterial taxa. Methane production by *M. smithii* represents the final step of inter-species hydrogen transfer, a syntrophic process in which hydrogen ($H_2$) and carbon dioxide ($CO_2$) released by fermentative bacteria are consumed by methanogens to yield methane and water[54]. By removing $H_2$, *M. smithii* enhances bacterial fermentation efficiency and indirectly supports metabolic pathways of anaerobes that are frequently enriched in CRC microbiomes. This interspecies hydrogen transfer and metabolic coupling could facilitate the growth of CRC-associated bacteria, such as *F. nucleatum*, *P. stomatis*, and *P. micra*, which rely on efficient fermentative metabolism under strictly anaerobic conditions[89,90]. In this context, methane acts not only as a metabolic by-product but also as an ecological marker and potential driver of microenvironments that favor CRC-associated taxa.

Given that *Methanobrevibacter* species rely on bacterial fermentation end products ($H_2$ and $CO_2$) for energy metabolism, their ecological function cannot be assessed individually. Therefore, we integrated archaeal and bacterial data to examine how *M. smithii* may influence CRC-associated bacterial taxa beyond methanogenesis and potentially contribute to colorectal carcinogenesis through microbe–microbe and potential host–microbe interactions.

While the loss of co-occurrence between commensal archaea and bacteria has been reported in CRC, *M. smithii* has shown positive co-occurrence associations with known CRC-associated taxa, such as *F. nucleatum*, *B. fragilis*, and *E. coli* in previous studies[22,35], a pattern we also observe by correlation analysis as well as presence/absence analysis in our study. Consistent with these associations, a tissue-based study detected archaea (including *M. smithii*) together with bacteria, such as *F. nucleatum* in CRC tumor biopsies[47].

These bacteria are known to promote carcinogenesis by inducing DNA damage, inflammation, and activating oncogenic pathways, suggesting that CRC-enriched archaea and bacteria may synergistically contribute to CRC[91]. On the other hand, previous studies have reported significant co-exclusion patterns between CRC-enriched archaea and CRC-depleted bacteria, including butyrate-producing species, such as *Clostridium beijerinckii* and *Clostridium kluyveri*, implying a potential antagonistic interaction in colorectal tumorigenesis[91].

An interesting finding of our study based on metabolic modeling was the universal export of succinate by all CRC-associated bacterial partners and its predicted uptake by *M. smithii*, highlighting succinate as a key cross-fed metabolite. Additionally, *M. smithii* was predicted to export riboflavin in all archaeal-bacterial pairings and to import several CRC-associated amino acids, such as asparagine, glutamine, leucine, and arginine. These metabolic exchanges suggest potential cooperative networks that could influence both microbial ecology and host disease processes, beyond $H_2$ exchange.

Experimental co-culture experiments validated our in silico findings. *M. smithii* growth was significantly enhanced only in the presence of *B. fragilis*, while *F. nucleatum* and *E. coli* showed archaeon-induced early growth acceleration. In these conditions, the archaeon did not show a significant growth increase in return, indicating only mild commensal benefits or fully unidirectional support.

In all co-cultures, succinate levels were higher than in mono-cultures, supporting predictions from genome-scale metabolic models and confirming succinate as a shared cross-feeding metabolite. However, the overall changes in metabolism differed between bacterial partners. The *F. nucleatum*–*M. smithii* pairing showed the most

extensive changes, with increased production of short-chain fatty acids (acetic, lactic, propionic, and butyric acids) and several amino acids (alanine, valine, isoleucine, leucine, phenylalanine, proline, and glycine), many of which have been linked to CRC[64,65,67].

Untargeted mass spectrometry-based metabolomics of supernatants further confirmed these findings, showing not only elevated levels of these CRC-related amino acids but also their modified forms, such as N-acetylated derivatives (e.g., N-acetyl-tyrosine, N-acetylvaline, and N6-acetyl-L-lysine).

Other detected metabolites are potentially directly connected to cancer-related pathways. For instance, arginine derivatives (N-α-acetyl-L-arginine, N-acetyl-arginine, and N-δ-acetylornithine) feed into nitric oxide and polyamine synthesis, both involved in CRC progression[67]. Interestingly, genes encoding cancer-related metabolites like polyamines have been shown to be more prevalent and diverse in gut and oral samples, affiliated with Euryarchaeota, especially methanogenic archaea[92]. Hypoxanthine, a purine compound known to affect gut function and tumor development, and I3P, an oncometabolite that supports tumor growth under tryptophan-starved conditions[63,72–74], were also found in the supernatant.

The co-cultures also showed significantly high abundance of some compounds likely impairing tumorigenesis. These compounds included γ-linolenic acid, which can trigger ferroptosis in tumor cells; 4E,8Z-sphingadiene, known to promote cancer cell death; apicidin F, a histone deacetylase inhibitor; and phenazine derivatives with strong anticancer activity[78,79,81–83]. Two of these compounds (γ-linolenic acid and 4E,8Z-sphingadiene) were significantly enriched in pure *M. smithii* cultures, indicating archaeal origin.

Our findings position *M. smithii* as a potentially active metabolic participant in CRC-associated microbial networks. Through both cooperative and asymmetrical metabolic interactions, *M. smithii* has the potential to modulate the abundance and function of key bacterial taxa and to actively contribute to a chemically rich microenvironment that includes metabolites with tumor-promoting and tumor-suppressive potential. Understanding how host factors like immune status affect the balance of these interactions, including the archaea-metabolome, will be key to determining their role in CRC development and therapy. Additionally, future in situ imaging and spatial-omics studies will be needed to validate these associations and clarify mechanistic roles.

## Methods
### Dataset collection
Relevant metagenomic investigations were identified through targeted keyword searches for diseases previously linked to gut methanogens[1–8,93], specifically CRC, IBD, T2D, MS, AD, SCZ, and PD (Supplementary Data 1). A systematic PubMed search was performed for English-language observational studies and meta-analyses published from 2000 until August 30, 2024. The search query used was: ("disease of interest") AND ((gut AND metagenome AND microbiome) OR fecal OR shotgun OR microbiota OR "whole genome"), with "disease of interest" encompassing CRC, IBD, PD, AD, T2D, MS, and SCZ. Additionally, reference lists from the identified articles and related meta-analyses were screened to identify additional studies. Additionally, the NCBI BioProjects database was screened for sequencing datasets using disease-specific terms (e.g., "CRC gut"), with filters applied for the "metagenome" data type and the human organism.

Studies employing shotgun metagenomic sequencing were considered. Only fecal shotgun metagenomic datasets were included, while studies with fewer than 20 case subjects or those involving participants under 18 years were excluded. Case subjects were defined as individuals with explicit disease diagnoses provided in the study metadata or NCBI BioProject records, whereas control subjects were those clearly described as healthy or designated as controls. Controls were defined as samples from individuals without a positive disease diagnosis. Thus,

it should be noted that controls do not necessarily present healthy individuals, but rather individuals without a specific diagnosis. Furthermore, studies were required to report metadata on sex, age, and/or BMI, or to have used case-control matched cohorts on at least two of these covariates. Studies that would have required additional ethical approvals or had restricted data access were omitted.

For CRC investigations, only samples from patients with confirmed CRC diagnoses were incorporated, excluding those with small or large adenomas. CRC staging information, based on the American Joint Committee on Cancer (AJCC) classification, was available for the Zeller et al.[41], Yachida et al.[45], Vogtmann et al.[46], Feng et al.[34], Wirbel et al.[42], and Gupta et al.[38] cohorts, which were therefore included in stage-specific analyses. In contrast, for the Hannigan et al.[94], Thomas et al.[37], and Yu et al.[33] datasets, staging information could not be identified from the available metadata. In the Yachida et al.[45] dataset, stage 0 samples were retained; however, in the other datasets, stage 0 groups comprised fewer than five samples or were not represented and were consequently excluded from stage-specific analyses in these cohorts. For the Wirbel et al.[42], Gupta et al.[38], and Feng et al.[34] datasets, the number of samples within individual stages fell below ten after case-control matching. To maintain statistical robustness, these datasets were analyzed only in overall case-control comparisons, without stage stratification. For analysis, stage I and II CRC cases were grouped as stage I/II, while stage III and IV CRC cases were grouped as stage III/IV. Consequently, CRC staging was taken into consideration for the Yachida et al.[45], Zeller et al.[41], and Vogtmann et al.[46] cohorts, where sufficient information and sample sizes were available. When matching CRC stages with controls, individual control samples were allowed to be reused across different stage groups. In the case of MS, inclusion was limited to two cohorts (from San Francisco, USA, and Edinburgh, Scotland) that consisted of treatment-naïve individuals and met the required sample size threshold, since treatment in these patients affects the abundance of the archaeome[26]. For AD, only pre-AD samples were considered, given the limited number of diagnosed cases. Sequence data were further limited to paired-end read libraries, thereby excluding datasets that combined single-end and paired-end reads. When available, sample covariates were extracted from the original study metadata, and for individuals with multiple SRA accessions, only the first available metagenome sample was used. Finally, any samples lacking accessible metadata from either the NCBI BioProject records or the published article were excluded.

Following these criteria, a total of 3243 samples from 19 studies across 12 countries were identified (Supplementary Table 1). Further details on study selection and the number of studies considered are provided in Fig. 1 and Supplementary Fig. 1.

## Data retrieval and processing

Raw sequencing data were retrieved from the NCBI Sequence Read Archive using SRA Toolkit (v3.1.0). Human-derived sequences were removed by aligning the raw reads to the GRCh38 human reference genome using *removehuman.sh* within the BBMap suite (v39.01) with default parameters. Read quality filtering, adapter trimming, base correction, and removal of low-complexity sequences were conducted concurrently using fastp (v0.23.4) with the following parameters: *--average_qual 20*, *--detect_adapter_for_pe*, *--correction*, *--overrepresentation_analysis*, and *--low_complexity_filter*. Unpaired reads resulting from trimming or filtering were corrected using BBMap (*repair.sh*), ensuring synchronization of read pairs for downstream analyses.

For microbial taxonomic profiling, we employed a Kraken2 and Bracken approach, which has been shown to outperform marker gene-based methods in recent benchmarking studies[95]. Taxonomic classification of the stool samples was carried out using Kraken2[96] (v2.1.2) with the unified human gastrointestinal genome (UHGG v2.0.1) database, which comprises a total of 4644 prokaryotic species, including 4616 bacterial species and 28 archaeal species, and enables identification of human gut archaeal species with high resolution[26,97,98]. For the CRC tumor and biopsy samples, the remaining metagenomic reads (i.e, after stringent filtering against the human reference genome (GRCh38) and quality control) were classified with Kraken2 using the Standard Database, which includes RefSeq bacterial, archaeal, and viral genomes, the human genome, and UniVec_Core sequences, to minimize false positives from host contamination. To improve classification accuracy and minimize incorrect lowest common ancestor (LCA) assignments, a confidence threshold of 0.3 was applied. Species-level abundance estimates were refined using Bracken (v2.7) (https://ccb.jhu.edu/software/bracken/) with read length 150. The resulting taxonomic profiles were merged into a comprehensive microbial abundance table for downstream analyses.

For diseases represented by multiple cohorts (CRC, pre-AD, and PD), datasets were subsequently analyzed in pooled form. For this purpose, batch effects across studies were mitigated using the adjust_batch() function from the MMUPHin (v1.23) R package[99], which applies a mixed-effects model to estimate and remove study-specific offsets. To ensure stringency, given the variability in sequencing depth, read length, preprocessing methods, sample collection, and DNA extraction protocols across publicly available datasets, each dataset was also processed independently with the same workflow, including identical quality control procedures, human-read removal, and taxonomic profiling.

## Archaeome and bacteriome community analyses

Shannon diversity indices were calculated separately for archaeal and bacterial communities. Species-level abundance tables were processed in R using the phyloseq package (v1.44.0). For each dataset, samples were rarefied to the 10th percentile of library sizes to normalize sequencing depth. Shannon diversity was estimated from rarefied data using the estimate_richness() function, and comparisons between case and control groups were performed using the Wilcoxon rank-sum test. To assess beta-diversity, principal coordinate analysis (PCoA) was performed based on Bray–Curtis distance on species-level abundance profiles after normalization of counts by total sum scaling. Sample clustering patterns were visualized in two-dimensional PCoA plots, with ellipses representing group dispersion (variability of samples within a group). To statistically evaluate differences in beta-diversity, we applied permuted multivariate analysis of variances (PERMANOVA) with 999 permutations.

## Machine learning analyses

Machine learning analyses were performed in Python (v3.10) using the *scikit*-learn (v1.5) and *shap* (v0.44) libraries. Prior to model training, all features were subjected to a three-step preprocessing pipeline to reduce noise and ensure comparability across features.

First, near-zero variance features were removed using the VarianceThreshold function (threshold $= 1 \times 10^{-5}$). Second, as feature values represented proportions ranging from 0 to 1, an arcsine square root transformation was applied to stabilize variance. Third, the resulting features were standardized (mean $= 0$, standard deviation $= 1$) using the StandardScaler function to ensure equal weighting in the model. The dataset was randomly partitioned into a training set (80%) and an independent validation set (20%).

A Random Forest classifier (RandomForestClassifier, scikit-learn) was trained using 2000 estimators with class weighting set to balanced and a fixed random seed (random_state $= 42$) to account for potential class imbalance and ensure reproducibility. Model performance was evaluated using stratified k-fold cross-validation (n_splits $= 5$, shuffle $=$ true), which preserves class distribution across folds. For each fold, a ROC curve was generated, and the mean $\pm$ standard deviation of the AUC was reported to summarize model performance stability.

Feature importance was estimated using two complementary approaches. First, permutation importance (permutation_importance, *scikit-learn*) was computed on the independent test set with 1000 repetitions (n_repeats = 1000, n_jobs = −1), providing the mean and standard deviation of the importance score for each feature. Second, SHAP (SHapley Additive exPlanations) values were computed to derive model-agnostic and directionally interpretable estimates of feature contributions. The top-ranked features were visualized based on mean permutation importance, and SHAP summary plots were used to confirm the robustness and interpretability of the model outputs.

## Selection of CRC bacterial markers

To identify CRC-associated taxa, we conducted a PubMed search using the keywords "CRC AND bacteria AND gut AND biomarkers" for studies published until August 30, 2024. After screening relevant publications and cross-referencing additional sources, we focused on bacterial CRC biomarkers at the species level. All identified bacterial taxa were incorporated into our metabolic modeling analyses alongside *Methanobrevibacter smithii* and CRC-associated bacterial markers.

## Metabolic modeling

Draft metabolic models were generated using gapseq (development version of 1.4.0, commit acb9647) on the genomes of all CRC biomarkers namely, *Segatella copri* (formerly *Prevotella copri*) (GenBank ID GCA_020735445.1), *Mediterraneibacter torques* (formerly *Ruminococcus torques)* (GenBank IDGCA_000210035.1), *Otoolea symbiosa* (formerly *Clostridium symbiosum* (GenBank ID GCA_008632235.1), *E. coli* (GenBank ID GCA_000025745.1), *Prevotella intermedia* (GenBank ID GCA_001953955.1), *Gemella morbillorum* (GenBank ID GCF_900476045.1), *Peptostreptococcus stomatis* (GenBank ID GCA_000147675.2), *Porphyromonas asaccharolytica* (GenBank ID GCA_000212375.1), *Akkermansia muciniphila* (GenBank ID GCA_009731575.1), *Fusobacterium nucleatum* (GenBank ID GCA_003019295.1), *Parviromonas micra* (GenBank ID GCA_900637905.1), *Bacteroides fragilis* (GenBank ID GCA_000025985.1), as well as *M. smithii* (GenBank ID GCA_000016525.1). The gut medium formulation used in this study followed previously established protocols[100,101]. Subsequently, the initial metabolic models were gap-filled utilizing this specified medium. To simulate co-culture interactions, community metabolic models were constructed for each pair of organisms using PyCoMo[50] version 0.2.7. The corresponding gut medium was applied to each model. The simulations included calculations of community growth dynamics, metabolite exchanges, and cross-feeding interactions between *M. smithii* and CRC-associated bacterial biomarkers, employing PyCoMo's default settings.

For the three CRC biomarkers with validated relevance[68,] *F. nucleatum*, *E. coli*, and *B. fragilis,* co-cultures with *M. smithii* were also simulated on relevant culture media. For this, in addition to the gut medium, the MS + BHI medium formulation was incorporated for gap-filling procedures and subsequent PyCoMo simulations, as previously detailed in our previous study[69] (full medium composition available in the referenced publication, and deposited also in the GitHub repository indicated in the Data availability section). Metabolite exchanges and cross-feeding patterns were computed by scaling the flux bounds from flux variability analysis based on the relative abundances of *M. smithii* and the CRC-associated bacteria, as determined by qPCR analysis performed on co-cultures. The resulting cross-feeding interaction profiles, highlighting metabolites produced by one organism and utilized by the other within each co-culture, were visualized using ScyNet[102] integrated into the Cytoscape[103] software (version 3.10.0).

## *E. coli* isolation and confirmation

To isolate the *E. coli* strain originally designated as *E. coli*_D based on the UHGG v2.0.1 taxonomy, and now classified as *E. coli* without the "_D" designation according to GTDB release 226 and subsequent updates (https://gtdb.ecogenomic.org/), we utilized a fecal sample from a previous study in which the co-occurrence of *M. smithii* and *E. coli*_D was confirmed through both cultivation and sequencing methods[69]. Specifically, 1 ml of the fecal sample from patient 51 (P51) of the aforementioned study was streaked onto CHROMagar™ *E. coli* medium. After incubation at 37 °C for 24 h, colonies displaying a characteristic intense blue coloration were selected for downstream processing.

Genomic DNA was extracted from the selected colonies using the PureLink™ Microbiome DNA Purification Kit (Invitrogen, Thermo Fisher Scientific, USA), following the manufacturer's protocol. To confirm the identity of the isolate as *E. coli*, two sets of primers were employed: one targeting the universal *E. coli* 16S rRNA gene (EC_F: 5′−CCAGGCAAAGAGTTTATGTTGA−3′, EC_R: 5′−GCTATTTCCTGCCGA TAAGAGA−3′)[104], and the other targeting the *adk* housekeeping gene (adkF: 5′−ATT CTG CTT GGC GCT CCG GG−3′, adkR: 5′−CCG TCA ACT TTC GCG TAT TT−3′)[105].

PCR amplification was performed as described previously[104], with small modifications. Briefly, PCR amplification of the *adk* gene was performed with an initial denaturation at 95 °C for 2 min, followed by 35 cycles of 95 °C for 1 min, 54 °C for 1 min, and 72 °C for 2 min. For the 16S rRNA gene, the protocol included an initial denaturation at 95 °C for 5 min, then 35 cycles of 92 °C for 1 min, 57 °C for 1 min, and 72 °C for 30 s. Both reactions concluded with a final extension at 72 °C for 5 min. PCR products of the expected size 212 bp for the 16S rRNA gene and 583 bp for *adk* were verified via 1.5% agarose gel electrophoresis. PCR products were subsequently purified using the Purification Kit (SolGent Co. Ltd., Daejeon, Republic of Korea), according to the manufacturer's instructions. The identity of the isolates was confirmed by direct Sanger sequencing of the purified PCR products.

## Co-culturing assays

We selected three CRC-associated bacterial strains, *F. nucleatum* (DSM 15643), enterotoxigenic *B. fragilis* (DSM 2151), and *E. coli* strain D (updated to *E. coli* in GTDB r226), for co-cultivation with *M. smithii* DSM 2375 (=ALI), due to their CRC-relevant virulence[68]. The experimental layout is illustrated in Supplementary Fig. 12.

Standard archaeal medium (MS medium) was prepared under anaerobic conditions as previously described[106]. Briefly, the MS culture medium was prepared per liter of distilled water and consisted of 0.45 g NaCl, 5 g NaHCO$_3$, 0.1 g MgSO$_4$·7H$_2$O, 0.225 g KH$_2$PO$_4$, 0.3 g K$_2$HPO$_4$·3H$_2$O, 0.225 g (NH$_4$)$_2$SO$_4$, and 0.060 g CaCl$_2$·2H$_2$O. In addition, 2 ml of a 0.1% (w/v) (NH$_4$)$_2$Ni(SO$_4$)$_2$ solution, 2 ml of a 0.1% (w/v) FeSO$_4$·7H$_2$O solution prepared in 0.1 M H$_2$SO$_4$, and 0.7 ml of a 0.1% (w/v) resazurin solution were included. The basal medium was supplemented with 1 ml each of 10× Wolfe's vitamin solution and 10× Wolfe's mineral solution.

Brain heart infusion (BHI) broth was prepared under anaerobic conditions in anaerobic chamber following standard protocols and flushed with nitrogen (100%). Subsequently, 10 ml of BHI broth and 20 ml of MS medium were combined in 100 ml serum bottles under anaerobic conditions. The medium was deoxygenated with nitrogen gas, supplemented with 0.75 g/l L-cysteine under anaerobic conditions, and adjusted to pH 7.0 if necessary. The 30 ml aliquots were sealed in 100-ml serum bottles with rubber stoppers and aluminum caps, pressurized with H$_2$/CO$_2$ (4:1), and sterilized by autoclaving at 121 °C for 20 min. Before use, 0.001 g/ml yeast extract and 0.001 g/ml sodium acetate were added under anaerobic conditions.

Time point designations were standardized relative to the initial inoculation of *M. smithii* ALI: t(−1) corresponds to the time of *M. smithii* ALI inoculation (0 h); t(0) denotes 24 h post−*M. smithii* ALI inoculation, or the point where bacterial mono-cultures in BHI + MS are initiated; t(0 + ) denotes the point bacterial strains are introduced to 24 h post−*M. smithii* ALI inoculation (initiation of co-cultures); t(1)

represents 48 h post−*M. smithii* ALI inoculation (24 h post-bacterial inoculation for co-cultures); and t(2) indicates 96 h post−*M. smithii* ALI inoculation (72 h post-bacterial inoculation).

Cultures of *M. smithii* ALI (5 ml) were initiated at t(−1) in BHI + MS medium for further co-culturing with a bacterium, to avoid the over-growth of bacteria due to higher incubation period for archaea and lower log phase time, with a parallel *M. smithii* ALI mono-culture pre-pared under identical conditions. The optical densities (OD) of *M. smithii* ALI master cultures in each experiment set used to inoculate in the mono- and co-cultures at t(−1) were: $0.07 \pm 0.007$ for *F. nucleatum* co-cultures, $0.05 \pm 0.008$ for *B. fragilis* co-cultures, and $0.47 \pm 0.09$ for *E. coli* co-cultures. At t(0), 0.2 ml of each bacterial strain pre-cultured overnight in BHI medium (*F. nucleatum* (OD = $0.82 \pm 0.05$), *B. fragilis* (OD = $0.90 \pm 0.04$), and *E. coli* (OD = $2.12 \pm 0.02$)), was separately inoculated into the *M. smithii* ALI culture to establish co-cultures, while bacterial mono-cultures were concurrently initiated in BHI + MS medium.

Optical density measurements were performed on 0.6 ml samples collected from *M. smithii* ALI mono-cultures and co-cultures at t(−1), t(0), t(1), and t(2), and from bacterial mono-cultures at t(0), t(1), and t(2). For metabolomic analyses, 1 ml samples were collected from *M. smithii* ALI mono-cultures, co-cultures, and bacterial mono-cultures at t(0), t(1), and t(2). For DNA extraction and subsequent qPCR analysis, 1 ml samples were collected from *M. smithii* ALI mono-cultures at t(2), from bacterial mono-cultures at t(1) and t(2), and from co-cultures at t(0), t(1), and t(2). At t(2), endpoint analyses were conducted on both co-cultures and *M. smithii* ALI mono-cultures to assess methane pro-duction, detect $F_{420}$-positive cells, and scanning electron microscopy was also conducted at t(2) for *M. smithii* ALI mono-cultures, co-cul-tures, and bacterial mono-cultures.

Blank BHI and BHI + MS media were incubated in triplicate for contamination monitoring, as well as for metabolomics experiments, while all inoculated cultures were performed in five biological repli-cates. All cultures were maintained at 37 °C with continuous shaking at 80 rpm throughout the experiments. DNA was extracted from cultures by using the PureLink™ Microbiome DNA Purification Kit (Invitrogen, Thermo Fisher Scientific, USA) according to the manufacturer's pro-tocols. DNA extracts were stored at −80 °C for further analysis. Culture samples for metabolomics were also stored at −80 °C until further analysis.

## CH₄ measurement and fluorescence microscopy

To evaluate whether methane (CH₄) production by *M. smithii* ALI was increased during co-cultivation, CH₄ concentrations were measured in the gas phase of the culture bottles with the CH₄ sensor BCP-CH4 (BlueSens gas sensor GmbH, Germany) following the manufacturer's instructions. Measurements were performed for both mono-cultures and co-cultures at the end of the incubation period.

Additionally, the presence and proliferation of *M. smithii* ALI were confirmed through the detection of the characteristic auto-fluorescence of coenzyme $F_{420}$ with maximum emission wavelength of 480[107]. Fluorescence imaging was performed using a Zeiss Axio Imager A1 microscope (Carl Zeiss AG, Germany) equipped with a fluorescence module. Observations were carried out using Zeiss filter set 05, com-prising a BP 395−440 excitation filter, an FT 460 beam splitter, and an LP 470 emission filter, in combination with a 100× Plan-NEOFLUAR objective.

## Scanning electron microscopy

The cell morphologies of the mono-cultures and co-cultures were examined using a Zeiss Sigma 500 V7P scanning electron microscope (Carl Zeiss AG, Germany). For sample preparation, 2 ml of culture (including supernatant) were immediately transferred on ice to the Core Facility Ultrastructure Analysis at the Medical University of Graz (Austria) for scanning electron microscopy (SEM), where cell pellets

were prepared by centrifugation at $4000 \times g$ for 10 min and processed as previously described[69]. Briefly, cells were deposited onto coverslips and chemically preserved in 0.1 M phosphate-buffered saline (pH 7.4) containing 2% paraformaldehyde and 2.5% glutaraldehyde. Samples were dehydrated through a graded ethanol series, followed by post-fixation with 1% osmium tetroxide for 1 h at room temperature, and subsequently subjected to a second ethanol dehydration sequence spanning 30–100% (vol/vol). Final drying was performed using hex-amethyldisilazane (HMDS), after which the coverslips were mounted on aluminum stubs with conductive double-sided carbon tape. Ima-ging was carried out on a Sigma 500 VP field-emission scanning elec-tron microscope (Zeiss, Oberkochen, Germany) equipped with a secondary electron detector, operating at an accelerating vol-tage of 5 kV.

## qPCR

Quantification of the absolute copy numbers of bacterial 16S rRNA and archaeal *mcrA* genes in the samples was conducted using a quantitative real-time PCR (qPCR) method with SYBR Green dye as previously described[9]. Briefly, each qPCR reaction mixture consisted of one microliter of DNA template combined with SYBR Green Supermix (Bio-Rad). The bacterial 16S rRNA genes were amplified using the primer pair 331F (5′-TCCTACGGGAGGCAGCAGT-3′) and 797R (5′-GGACTAC-CAGGGTATCTAATCCTGTT-3′)[108]. The thermal cycling protocol for bacterial amplification involved an initial denaturation step at 95 °C for 15 s, followed by 40 cycles consisting of denaturation at 94 °C for 15 s, annealing at 54 °C for 30 s, and extension at 73 °C for 40 s. For the detection and quantification of *M. smithii* ALI, primers M1F (5′-GCAATGCAAATTGGTATGTC-3′) and M1R (5′-TCATTGCGTAGT-TAGGRTAGT-3′) were used, specifically targeting the *mcrA* gene (sin-gle-copy gene). The thermal cycling conditions included an initial denaturation at 94 °C for 3 s, followed by 40 cycles of denaturation at 94 °C for 45 s, annealing at 56 °C for 45 s, and elongation at 72 °C for 30 s.

The quantification cycle (Cq) values were analyzed using the regression method available in Bio-Rad CFX Manager Software (ver-sion 3.1). The absolute gene copy numbers for bacterial 16S rRNA and methanogenic *mcrA* genes were determined based on Cq values and corresponding amplification efficiencies derived from standard curves. These standard curves were established according to pre-viously described protocols[9] based on standard curves from defined DNA samples of *E. coli* for bacterial 16S rRNA genes and the *mcrA* gene from the *M. smithii*[109,110].16S rRNA gene copy numbers were normalized using species-specific values from the rrnDB database (*F. nucleatum* = 5 copies, *B. fragilis* = 6 copies, and *E. coli* = 7 copies). The detection thresholds were established using the mean Cq values obtained from non-template control reactions. All qPCR assays were performed in three technical replicates, with each assay replicated across five inde-pendent biological replicates of both mono and co-cultures.

## Metabolite quantification using nuclear magnetic resonance

In order to gain initial insights into the consumption/production of amino acids, the metabolic activity of *M. smithii* ALI was investigated by nuclear magnetic resonance (NMR) spectroscopy using *M. smithii* ALI cultures previously described in an earlier study[106]. Measurements were performed in triplicate for *M. smithii* ALI at 72, 168, and 240 h post-inoculation in MS medium supplemented with yeast extract, fol-lowing the experimental protocol outlined in the same reference.

Metabolomic profiling of mono- and co-culture samples (cell + supernatant) collected for this study at different timepoints (Supple-mentary Fig. 12) was also conducted through NMR spectroscopy. For this purpose, samples were initially treated using protein precipitation by the addition of methanol to give a methanol−water mixture at a 2:1 ratio. After centrifugation, supernatants were collected and subse-quently dried by lyophilization. The lyophilized extracts were then

reconstituted in a sodium phosphate-buffered solution supplemented with an internal NMR reference standard, 4.6 mM 3-trimethylsilyl propionic acid-2,2,3,3-d4 sodium salt, before being transferred into NMR sample tubes.

Spectral acquisition was performed using a Bruker Avance Neo 600 MHz spectrometer, equipped with a triple resonance inverse probe, operated at a controlled temperature of 310 K. Data were captured utilizing the Carr–Purcell–Meiboom–Gill pulse sequence over 128 scans, with acquisition and initial processing conducted via Topsin 4.5 software (Bruker GmbH, Rheinstetten, Germany). Post-acquisition data treatment, involving spectral alignment and probabilistic quantile normalization (PQN), was executed using MATLAB software (version 2014b, MathWorks, Natick, MA, USA).

For the precise quantification of metabolites displaying significant enhancement in signal intensities within co-cultured samples compared to mono-culture controls, integration of targeted metabolite peaks from the aligned spectra was conducted following baseline correction utilizing trapezoidal integration methods. Subsequent normalization against the proton number, specific J-coupling characteristics, and the integral of the internal standard allowed for accurate determination of metabolite molar concentrations.

## Metabolite determination using mass spectrometry

The blank MS + BHI medium ($n = 3$ replicates), as well as supernatant of the co-culture of *M. smithii* ALI and *F. nucleatum* ($n = 3$ replicates) were measured using the Elute PLUS LC system (Bruker, Bremen, Germany) coupled to a timsTOF Pro 2 mass spectrometer (Bruker, Bremen, Germany) with a vacuum-insulated probe heated electrospray ionization (VIP-HESI) source in both reverse phase (RP) and hydrophilic interaction (HILIC) modes. System suitability of the LC-MS setup was confirmed by use of weekly measurements of the QSee Performance Test setup from Bruker (Bremen, Germany), using a mixture of 8 synthetic compounds.

A pooled QC sample was prepared by combining 100 μL aliquots of each supernatant. The pooled QC was measured intercalating between samples to perform within-batch correction.

RP separations were conducted on an Intensity Solo 2 C18 Column (100 Å; 2.0 μm; 2.1 mm × 100 mm; #BRHSC18022100, Bruker) with 0.1% formic acid (ROTIPURAN® ≥99%, LC-MS Grade, Carl Roth, Karlsruhe, Germany) in MilliQ water as mobile phase A and 0.1% formic acid in 9:1 (v:v) acetonitrile: MQ water (≥99.9%, HiPerSolv CHROMANORM® for LC-MS, VWR, Darmstadt, Germany) as mobile phase B. A 5 μl injection of each sample was used. The separation was carried out at a flow rate of 0.6 ml/min with a column temperature maintained at 50 °C. The following gradient was applied: 0–2 min, 5% B; 2–10 min, 5-40% B; 10–11 min, 40-98% B; 11–13 min, 98% B; 13–13.1 min, 98-5% B; 13.1–15.5 min, 5% B.

HILIC separations were performed on an ACQUITY UPLC BEH Amide column (130 Å, 1.7 μm, 2.1 mm × 150 mm; #186004802, Waters) with 10 mM ammonium formate and 0.1% formic acid in MilliQ water as mobile phase A and 10 mM ammonium formate and 0.1% formic acid in acetonitrile as mobile phase B. Injection volume was set to 5 μl, the flow rate of 0.5 ml/min and the column temperature at 40 °C. The gradient was as follows: 0–1 min, 100% B; 1–6 min, 100–90% B; 6–10 min, 90–75% B; 10–11 min, 75–60% B; 11–12 min, 60% B; 12–12.1 min, 50–100% B; 12.1–21 min, 100% B.

The VIP HESI source was set to default conditions: endplate offset 500 V; capillary 4500 V; nebulizer gas 2.0 bar; dry gas 8.0 l/min; dry temp 230 °C; sheath gas 4.0 l/min; sheath gas temperature 400 °C. The probe head was put at minimum distance to the front-end assembly for maximum intensity. LC-MS/MS data were acquired in both positive and negative DDA-PASEF modes, for a mass range of m/z 20–1300. Default Bruker PASEF acquisition parameters for MS/MS acquisition were used: 2 ramps (12 precursors each) per cycle; resulting cycle time 0.69 s; Intensity threshold 100 counts; target Intensity 4000 counts

(signals below that threshold will be scheduled for MS/MS fragmentation more often); active exclusion activated (0.1 min; reconsider if intensity increase is at least 2-fold). Data acquisition was performed using Bruker software timsControl® and Compass HyStar® software. Quality control (QC) samples were run every ten injections in HILIC mode, and every five injections in RP mode, and blank samples were analyzed at the beginning and the end of each batch using $H_2O$ for RP and methanol for HILIC.

Raw data were processed using MetaboScape® (version 2024b, Bruker, RRID:SCR_026044) with four-dimensional (4D) feature extraction, capturing mass-to-charge ratio (m/z), isotopic pattern quality, retention times, MS/MS spectra, and collision cross-section (CCS) values. Feature extraction was performed using the T-ReX® 4D algorithm (RRID:SCR_026044), followed by annotation through the Bruker Human Metabolome Database (HMDB, RRID:SCR_007712) and the NIST Mass Spectral Library (RRID:SCR_014668) as well as additional "target lists" derived from several CCS databases (Pacific Northwest National Laboratory, METLIN, MiMeDB), resulting in Level 2 annotation according to Sumner et al.[111]. Matching was performed using four parameters: m/z (accurate mass, error <5 ppm), mSigma (isotopic pattern, score <100), MS/MS match (score >600) and CCS accuracy (error <3%).

High-quality annotations for final analysis were selected through visual inspection of chromatogram and ion mobilogram peak quality and sufficient annotation scores (match for at least 2 of the 4 parameters). Duplicate annotations were resolved by removing annotations with (1) more missing values, (2) lower overall annotation quality and (3) lower CCS accuracy until only unique annotation remained.

Raw metabolite concentrations of 489 compounds (level 2 annotation) were processed with MetaboAnalyst Version 6 (www.metaboanalyst.ca). Missing values were replaced by LoDs (1/5 of the minimum positive value of each variable). Samples were normalized by PQN to account for different cell counts and dilution effects and log transformed (base 10).

## Statistical analysis

All statistical analyses were conducted using R (version 4.3.1) in RStudio (version 2023.06.1 + 524). In datasets where case-control samples were not initially matched for potential confounding variables (age, sex, and BMI), additional control for these factors was applied where feasible. Specifically, case-control samples were matched within each dataset by sex, age (±5 years), and BMI (±3 units) to reduce potential bias (Supplementary Data 1).

Subsequent differential abundance analyses for both bacteria and archaea were carried out following centered log-ratio (CLR) transformation of the dataset, using the Wilcoxon rank-sum test, followed by Benjamini–Hochberg false discovery rate (FDR) correction for multiple testing. Following previously established criteria[112], provided that more than two datasets were available, a taxon was associated with a given disease if it demonstrated a statistically significant association ($p$-adjusted <0.05) in the same direction across at least two independent studies of the disease. Co-abundance networks of *M. smithii* with CRC-associated bacterial taxa in case samples were inferred using Spearman's rank correlation, and false discovery rate (FDR) correction was applied using the Benjamini–Hochberg method. Taxa were retained for analysis only if they were detected in at least 5% of case samples. Presence-absence matrices of *M. smithii* with CRC-associated bacterial taxa were generated from abundance tables by classifying values > 0 as "present." For each taxon and within each group (control or CRC), $2 \times 2$ contingency tables between *M. smithii* and the bacterial taxon were analyzed using Fisher's exact test to estimate odds ratios (OR), 95% confidence intervals, and Benjamini–Hochberg-adjusted $p$-values.

Two-sided Welch's t-test and two-sided Wilcoxon rank-sum test were used to compare co-cultures and mono-cultures across individual NMR metabolomic profiles, qPCR data, and methane measurements

based on the normal distribution of data. For NMR metabolomic data, multiple testing correction was applied using the Benjamini–Hochberg FDR method. Significant changes in metabolites measured with mass spectrometry in the co-culture of *M. smithii* ALI and *F. nucleatum* compared to the blank medium were determined by t-test with a threshold of *p*-adjusted <0.05 and minimum fold change threshold of FC > 1.5.

## Reporting summary

Further information on research design is available in the Nature Portfolio Reporting Summary linked to this article.

## Data availability

Raw sequencing data for stool samples All sequencing data analyzed in this study were obtained from previously published datasets and are publicly available from the European Nucleotide Archive (ENA) and the NCBI BioProject/SRA databases under the following accession numbers: PRJDB4176, PRJEB6070, PRJEB7774, PRJEB10878, PRJNA389927, PRJEB12449, PRJEB27928, PRJNA447983, PRJNA531273 and PRJNA397112, PRJNA400072, SRA045646 and SRA050230 [https://www.ncbi.nlm.nih.gov/sra/SRA050230], PRJEB32762, PRJEB47976, PRJNA798058, PRJEB29127, PRJNA834801, PRJNA743718, PRJEB53401, and PRJEB17784. The NMR raw data generated in this study are available in Zenodo at https://doi.org/10.5281/zenodo.16311518. The LC-MS raw data generated in this study are available in Zenodo at https://doi.org/10.5281/zenodo.16367666. All additional data generated and analyzed in this study, including Kraken2/Bracken outputs, gapseq outputs (including genome-scale metabolic models, gap-filled models, reaction and gene annotations, as well as pathway and transporter predictions), PyCoMo outputs (including flux variability analyses, metabolite secretion and uptake predictions, and community) are publicly available in our GitHub repository (https://github.com/CME-lab-research/archaeome-disease-profiling/). Source data are provided with this paper.

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

## Acknowledgements

This research was funded in whole or in part by the Austrian Science Fund (FWF) [10.55776/P32697 (given to C.M.E.), excellence cluster "Microbiomes Drive Planetary Health" 10.55776/CoE7 (C.M.E., C.D., and G.G), and SFB ImmunoMetabolism 10.55776/F8300 (C.M.E.)]. C.M.E. has

received funding from the European Research Council (ERC) under the Horizon Europe research and innovation programme (Project ID 101199346, ERC-2024-ADG). T.M. is grateful to the Austrian Science Fund (FWF) for excellence cluster 10.55776/COE14, grants DOI 10.55776/P28854, 10.55776/I3792, 10.55776/DOC130, and 10.55776/W1226, the Austrian Research Promotion Agency (FFG) grants 864690 and 870454; the Integrative Metabolism Research Center Graz; the Austrian Infrastructure Program 2016/2017; the Styrian Government (Zukunftsfonds, doc.fund program); the City of Graz; and BioTechMed-Graz (flagship project). This project was funded in part by the FFG and the European Union (EFRE) under grant 912192. T.M. and H.H. acknowledge the Center for Medical Research for laboratory access. C.T. reports a research grant by Bruker Switzerland AG. For open access purposes, the author has applied a CC BY public copyright license to any author-accepted manuscript version arising from this submission. We gratefully acknowledge the computational resources provided by the MedBio-Node at the Medical University of Graz, funded by the Austrian Federal Ministry of Education, Science, and Research through the Hochschulrat-Struktur Mittel 2016 grant within BioTechMed Graz. We also thank the ZMF Core Facility Computational Bioanalytics team at the Medical University of Graz for their support. We thank Claire Lamb for assistance with the provision of the *Bacteroides fragilis* strain. We thank Charlotte Neumann for her help in creating some illustrations for this study. R.M. was supported by the local PhD program MolMed.

## Author contributions

R.M. designed the study, collected data, performed bioinformatics, data analysis, plotting, and drafted the manuscript. A.M. co-designed the study. T.Z. and L.W. performed sample cultivation. C.K. performed qPCR. K.F. assisted with plot preparation. P.M. performed *E. coli* isolation and confirmation. H.H. and T.M. performed NMR metabolomics. J.S. and C.T. performed MS metabolomics. D.P., K.H., and D.K. performed SEM. M.D performed the machine learning analysis. C.D. helped with data analysis. G.G. and A.L. assisted with sample preparation. G.G., C.T., C.D., and A.L. commented on and revised the manuscript. C.M-E. designed and supervised the study, and drafted and revised the manuscript. All authors reviewed and approved the final manuscript.

## Competing interests

The authors declare no competing interests.

## Additional information

**Supplementary information** The online version contains Supplementary material available at https://doi.org/10.1038/s41467-026-69711-7.

