## [Transparent Peer Review File · Nature Communications]

Cross-domain metabolic interactions link *Methanobrevibacter smithii* to colorectal cancer microbial ecosystems

Corresponding Author: Professor Christine Moissl-Eichinger

Version 0:

Reviewer comments:

Reviewer #1

(Remarks to the Author)

The paper by Mohammadzadeh et al. presents a comprehensive meta-analysis of the human gut archaeome across multiple diseases, utilizing high-quality shotgun metagenomes from 1,882 fecal samples collected in 19 studies across 12 countries. The datasets include different disorders, such as colorectal cancer (CRC), type 2 diabetes (T2D), inflammatory bowel disease (IBD, including Crohn's disease (CD) and ulcerative colitis (UC)), multiple sclerosis (MS), Alzheimer's disease (AD), schizophrenia (SCZ), and Parkinson's disease (PD). Although associations between Archaea and medical disorders are common yet highly variable, the overall trend indicates an enrichment of the methanogen *Methanobrevibacter smithii* in CRC patients compared to healthy individuals. Metabolic modelling, in vitro co-culture experiments with three CRC biomarkers with validated relevance (*F. nucleatum*, *E. coli*, and *B. fragilis*) and metabolomic analyses identified archaeal and bacterial-derived compounds, with succinate and amino acids as key mediators of archaeal-bacterial metabolic exchanges. In addition, archeal metabolites (γ -linolenic acid and 4E,8Z-sphingadiene) that could directly interact with host receptors and may have anticancer activity were identified.

In general, this work is of interest to readers working in human gut microbiome research, archeome and colorectal cancer. The experiments are well-designed, logical and comprehensive, leaving little doubt that the primary conclusions are correct. The overall writing is comprehensive and insightful. I have a few minor queries that authors may wish to consider.

Title and abstract L39: overall, this paper provides evidence of metabolic interactions (exchange of metabolites) between *Methanobrevibacter smithii* and three important bacteria associated with CRC. I recommend replacing "trophic control" with "metabolic interactions", as the study does not present evidence for regulatory mechanisms operating across different trophic levels.

L152-154: This sentence is unclear. Could you please clarify what is meant by the "standardized workflow" mentioned here?

L203-210: this section gives the impression that the *E. coli* strain was isolated in the present study. Some details can be omitted as already described in the previous paper.

L708-712: The authors claim that *M. smithii* may export riboflavin and benefit its bacterial partners. Could the authors clarify the basis for this claim? Do these bacteria possess transporters for riboflavin uptake?

L741: this should read -> Colors indicate metabolite export or production (yellow).

L742: this should read -> Only metabolites involved in cross-feeding interactions are shown.

L771-776: This section reads as somewhat repetitive; I suggest deleting it.

L779-791: The figure could be moved to the supplementary materials as it depicts what is already described in the methods. Also, why not use the specific bacterial names in the figure? Alternatively, label them as "Bacterium 1," "Bacterium 2," and "Bacterium 3" for clarity.

L810-813: This statement appears counterintuitive; if active cross-feeding is occurring, succinate should not accumulate to high levels in the co-cultures. As the measurements at T1 and T2 were taken once the co-culture had reached the stationary phase (see Figure S9A), it may be more informative to assess succinate levels during the exponential phase of growth of the coculture.

L888: export (green) -> export (yellow).

Reviewer #2

(Remarks to the Author)

In this study, the authors focused on the Archaea in human disease, especially colorectal cancer. Furthermore, the metabolic

modelling approach can estimate the relationship between *Methanobrevibacter smithii* and CRC-causing bacteria, and is validated by the in vitro coculture approach. These works are excellent; however, additional computational analysis is needed to support the idea that *M. smithii* communicates with CRC-causing bacteria.

1. Regarding the heatmap, especially Figure 4, the color of 'the amount of metabolites' may be changed to prevent misunderstanding. I suggested that a minus value may be indicated by the blue color.
2. CRC-causing bacteria were chosen by a literature review. Could you check the bacterial interaction between *M. smithii* and CRC-causing bacteria by correlation analysis? These computational results may support the result of the biological experiment.
3. I am not sure the the specificity of the consumption of the compounds by *M. smithii*. For example, there are many bacteria that have amino acid degradation genes. Could you check the co-occurrence between amino acids and these bacteria by network analysis?
4. Methane is one of the specific compounds produced by *M. smithii*. What do you think of the potential that *M. smithii* interacts with CRC-causing bacteria by producing Methane?
5. A previous study from Yachida et al reported that *M. smithii* was enriched in the late CRC; on the other hand, *F. nucleatum* was increased along with CRC progression. In this study, you validated the interaction of *M. smithii* and *F. nucleatum* by biological experiment; however, this interaction may not be observed in the cohort study. This is one of the limitations of this study.

Reviewer #3

(Remarks to the Author)

The authors have carried out a literature review and meta analysis of selected human gut microbiome studies to try to strengthen their claim that archaea, specifically *Methanobrevibacter smithii*, is a controlling contributor to a human disease. Of the thousands of studies, both large and small, that have been carried out on humans across the world from all age groups, they focus on twenty, two of which show a statistically significant correlation between disease and enriched archaeal species, however the effect on disease severity is not discussed. The correlation studies would be strengthened by including machine learning tools for multifactorial analysis to capture a more comprehensive picture of archaeal contribution to microbiome-associated digestive diseases, as is the standard in the field.

The authors then seek to bolster the claim that archaea promote colorectal cancer by culturing them with known cancer-promoting bacteria and characterizing metabolites in vitro. Unfortunately there are systematic methodological choices made by the investigators that affect the data interpretations. Rigorous culturing and phenomics experiments are needed to conclude how archaea interact with cancer-promoting bacteria, however the authors focus on one set of experiments that require enrichment of archaea to begin with – likely skewing the outcomes and thus overplaying the interpretations with respect to disease contribution. Evidence of physical co-association between archaea and cancer-promoting bacteria are lacking, making it difficult to understand how secretion and consumption of pro- and anti-cancer metabolites by the microbes in a crowded gut environment would contribute to establishing disease. The manuscript would be strengthened by accounting for the mechanisms of cancer development caused by bacteria, similar to how other researchers have shown that archaea promote the impact of frank pathogens on health.

Version 1:

Reviewer comments:

Reviewer #1

(Remarks to the Author)

The authors have satisfactorily addressed all my comments. The revised version reads very well and represents a valuable contribution to the field. I wish to congratulate the authors on their excellent work and the effort invested in improving this study.

Reviewer #2

(Remarks to the Author)

In this revision, the authors conducted further analysis to elucidate the association between *M. smithii* and *F. nucleatum* across various stages of CRC by co-correlation and machine learning approaches. Furthermore, the author clearly explains the potential metabolic cross-feeding between *M. smithii* and CRC-associated bacteria, as well as the potential role of methane in interactions between host and bacteria. These works are great and equivalent to accepting.

Reviewer #4

(Remarks to the Author)

I have been asked by Nature Comm to review the response to reviewers for this manuscript. I read with great interest the manuscript examining shotgun sequencing data from a number of studies pooled to examine the potential relationship between methanogens and colorectal cancer. The authors describe the association between methanogens and CRC cases in case-control studies and then go on to discuss the effects on the bacterial microbiome. Specifically, the authors find that methanogens enhance or appear to be associated with an increase in bacterial players that have known pathogenic implications in progression to CRC such as fusobacterium, E. coli, and others. Co-culture studies were helpful in showing that methanogens could encourage the growth of these other organisms in simplistic growth simulations. They further go on to study the supernatant for these experiments seeming to find some biochemical components that have published relationships to mucosal health and CRC development as well.

Overall, the authors received 3 reviews with two of the three reviews having minor commentary and corrections which the authors handled well. The final reviewer was more concerned about the overreaching arguments. Specifically, there was a concern about the use on only a few studies of the many published over time. The authors appear to sufficiently address the reasons for their selection of databases (primarily finding databases that use shotgun sequencing sufficient enough to assess for archaea).

MOre importantly, the reviewer appeared to be suggesting a more guarded tone to the association with CRC. The study is not a cause and effect study. It is an association study. Methanogens are more associated with CRC. The presence of methanogens appear to draw or encourage a greater diversity and consortium of bacteria some of whom have associations with CRC. They prove the encouragement of growth in co-culture and some of the byproducts of their co-culture were interesting molecules that could have importance based on the previous literature in CRC development. But all of this is associative and not cause and effect.

I think the paper is very important and warrants publication. I think the author have toned down any language around cause and effect. I do believe they could do a bit more to tone this down. This paper does not say that methanogens cause cancer and hopefully is not reading in that way. In fact, data suggest methanogens may be associated with healthy aging and maybe even longevity. But, it could be strain dependent for methanogens as well as epigenetic interrelationships that are yet to be determined. Thus being introspective until the next round of data can draw more direct lines would be importang.

Dear editor and reviewers,

We sincerely appreciate the valuable comments provided on our manuscript. In response to these, we have undertaken a comprehensive revision of the manuscript, in which we address all the raised concerns. Below, we provide a detailed summary of the revisions made in response to the reviewers' comments.

Reviewer #1 (Remarks to the Author):

The paper by Mohammadzadeh et al. presents a comprehensive meta-analysis of the human gut archaeome across multiple diseases, utilizing high-quality shotgun metagenomes from 1,882 fecal samples collected in 19 studies across 12 countries. The datasets include different disorders, such as colorectal cancer (CRC), type 2 diabetes (T2D), inflammatory bowel disease (IBD, including Crohn's disease (CD) and ulcerative colitis (UC)), multiple sclerosis (MS), Alzheimer's disease (AD), schizophrenia (SCZ), and Parkinson's disease (PD). Although associations between Archaea and medical disorders are common yet highly variable, the overall trend indicates an enrichment of the methanogen *Methanobrevibacter smithii* in CRC patients compared to healthy individuals. Metabolic modelling, in vitro co-culture experiments with three CRC biomarkers with validated relevance (*F. nucleatum*, *E. coli*, and *B. fragilis*) and metabolomic analyses identified archaeal and bacterial-derived compounds, with succinate and amino acids as key mediators of archaeal-bacterial metabolic exchanges. In addition, archeal metabolites (γ -linolenic acid and 4E,8Z-sphingadiene) that could directly interact with host receptors and may have anticancer activity were identified. In general, this work is of interest to readers working in human gut microbiome research, archeome and colorectal cancer. The experiments are well-designed, logical and comprehensive, leaving little doubt that the primary conclusions are correct. The overall writing is comprehensive and insightful. I have a few minor queries that authors may wish to consider.

We appreciate the overall positive feedback of the reviewer.

Title and abstract L39: overall, this paper provides evidence of metabolic interactions (exchange of metabolites) between *Methanobrevibacter smithii* and three important bacteria associated with CRC. I recommend replacing “trophic control” with “metabolic interactions”, as the study does not present evidence for regulatory mechanisms operating across different trophic levels.

We appreciate your constructive suggestion and have revised the title and abstract accordingly. The updated title now is: “*Methanobrevibacter smithii* associates with colorectal cancer through

metabolic interaction with the cancer bacteriome”, and the last sentence in the abstract was changed: “This provides the first mechanistic link between human gut archaeome and CRC and highlights its role in modulating health in humans through metabolic interaction with the resident bacteriome.”

L152-154: This sentence is unclear. Could you please clarify what is meant by the “standardized workflow” mentioned here?

Thank you for pointing this out. We have clarified the phrasing by replacing “standardized workflow” with “the same workflow, including identical quality control procedures, human-read removal, and taxonomic profiling” which more accurately reflects our intended meaning.

L203-210: this section gives the impression that the *E. coli* strain was isolated in the present study. Some details can be omitted as already described in the previous paper.

Thank you for this comment. To clarify, the *E. coli* strain was indeed isolated in the present study. In the previous work we cited, the strain had only been identified through shotgun sequencing, but it was not isolated at that time.

No change has been made to the manuscript.

L708-712: The authors claim that *M. smithii* may export riboflavin and benefit its bacterial partners. Could the authors clarify the basis for this claim? Do these bacteria possess transporters for riboflavin uptake?

Thank you for this comment. Riboflavin transporters, along with transporters for other metabolites, were predicted from the genomes of the microbial taxa using the *gapseq* tool (Zimmermann et al., 2021; <https://doi.org/10.1186/s13059-021-02295-1>). Specifically, *gapseq* identified transporters belonging to the TC family 2.A.1.3.72 with high probability, which are annotated in the Transporter Classification Database (TCDB) as riboflavin transporters (RibZ). These predictions suggest that *M. smithii* possesses the genetic potential for riboflavin transport. However, we acknowledge that these predictions are based on computational inference and have not yet been experimentally validated.

To ensure transparency, the list of all predicted transporters have been deposited in our public GitHub repository (https://github.com/CME-lab-research/archaeome-disease-profiling/tree/main/02_gapseq_output), where the files with the suffix “-Transporter.tbl” lists the corresponding annotations for each taxon. To clarify this in the manuscript, we have added a

sentence in the *Data and Code Availability* section in lines 1093-1094 (see “transporter file in *gapseq* output of *M. smithii*” in the Data and Code Availability section).

L741: this should read -> Colors indicate metabolite export or production (yellow).

Thank you for pointing this out. We corrected the sentence.

L742: this should read -> Only metabolites involved in cross-feeding interactions are shown.

Thank you for pointing this out. We corrected the sentence.

L771-776: This section reads as somewhat repetitive; I suggest deleting it.

Thank you for pointing this out. We deleted the paragraph.

L779-791: The figure could be moved to the supplementary materials as it depicts what is already described in the methods. Also, why not use the specific bacterial names in the figure? Alternatively, label them as “Bacterium 1,” “Bacterium 2,” and “Bacterium 3” for clarity.

Thank you for this helpful suggestion. We moved the Figure to the Supplementary Materials (Supplementary Fig. S12). Regarding the labeling, we have revised the figure to use the “Bacterium 1,” “Bacterium 2,” and “Bacterium 3” to improve clarity.

L810-813: This statement appears counterintuitive; if active cross-feeding is occurring, succinate should not accumulate to high levels in the co-cultures. As the measurements at T1 and T2 were taken once the co-culture had reached the stationary phase (see Figure S9A), it may be more informative to assess succinate levels during the exponential phase of growth of the coculture.

Thank you for this great comment. We agree that if succinate cross-feeding were fully balanced between production and consumption, one would not expect substantial succinate accumulation in co-cultures.

However, the measured succinate levels likely represent the net balance between production and consumption at the time of sampling, rather than the instantaneous flux of cross-feeding during exponential growth.

As the co-cultures had reached stationary phase, it is plausible that the producer strain continues to generate succinate at a higher rate or for a longer period than the consumer could utilize, resulting in residual accumulation. This interpretation is consistent with our metabolomics data, which indicate a general increase in succinate abundance in co-cultures compared to the corresponding monocultures. Indeed, time-resolved metabolite profiling during the exponential phase would provide further insights into the dynamics of succinate cross-feeding and we will integrate this experiment, along with stable isotope probing, into our future studies.

No changes were made to the manuscript.

L888: export (green) -> export (yellow).

Thank you for pointing this out. We corrected it.

Reviewer #2 (Remarks to the Author):

In this study, the authors focused on the Archaea in human disease, especially colorectal cancer. Furthermore, the metabolic modelling approach can estimate the relationship between *Methanobrevibacter smithii* and CRC-causing bacteria, and is validated by the in vitro coculture approach. These works are excellent; however, additional computational analysis is needed to support the idea that *M. smithii* communicates with CRC-causing bacteria.

1. Regarding the heatmap, especially Figure 4, the color of 'the amount of metabolites' may be changed to prevent misunderstanding. I suggested that a minus value may be indicated by the blue color.

Thank you for this helpful suggestion. We changed the heatmap colors as suggested.

2. CRC-causing bacteria were chosen by a literature review. Could you check the bacterial interaction between *M. smithii* and CRC-causing bacteria by correlation analysis? These computational results may support the result of the biological experiment.

Thank you for this helpful suggestion. We checked the correlation of CRC-causing bacteria and *M. smithii* both in global data and each study independently (see new Supplementary Fig. 9) and co-occurrence analysis in global data based on presence-absence pattern (see new Supplementary Fig. 10) (Lines 334 and 346).

3. I am not sure the the specificity of the consumption of the compounds by *M. smithii*. For example, there are many bacteria that have amino acid degradation genes. Could you check the co-occurrence between amino acids and these bacteria by network analysis?

We appreciate the reviewer's feedback. We would like to clarify that this specific analysis was not intended to assess the exclusivity of substrate utilization, but was rather performed to identify potential metabolic cross-feeding between *M. smithii* and CRC-associated bacteria. We fully agree that many gut microorganisms possess amino acid degradation pathways and likely contribute to the broader gastrointestinal metabolic network. The co-occurrence of amino acids with (CRC-associated) bacteria is indeed a valid point and has been addressed in previous studies, summarized for example by Wu *et al.* (2024; <https://doi.org/10.1016/j.biopha.2024.116410>). Our approach therefore focused on highlighting putative metabolic connections involving *M. smithii*, rather than implying that these substrates are uniquely consumed/ produced by this archaeon. A sentence has been added in Lines 399-401, to clarify this also for the readers.

4. Methane is one of the specific compounds produced by *M. smithii*. What do you think of the potential that *M. smithii* interacts with CRC-causing bacteria by producing Methane?

Thank you for this question! The potential impact of *Methanobrevibacter smithii* on CRC-associated bacteria via methane production can be considered from two perspectives: (i) the process of methanogenesis itself, and (ii) the biological role of methane as a molecule.

Regarding methanogenesis, we have expanded the discussion (Lines 592-602) to include its ecological implications. Methanogens such as *M. smithii* efficiently remove excess hydrogen produced by bacterial fermentation, thereby maintaining low hydrogen partial pressure and promoting the thermodynamic feasibility of various bacterial metabolic pathways. This hydrogen-scavenging activity can indirectly influence the growth and metabolism of CRC-associated bacteria. But, as we show in our manuscript, this interaction goes way beyond mere hydrogen transfer.

In contrast, the direct biological role of methane is less clear. Most gut microorganisms cannot utilize methane, as methanotrophy requires oxygen or oxygenated electron acceptors - both not typically present in the human gut. Also, methanotrophs have not been reported in this environment. Nonetheless, methane itself could serve as a signaling molecule, influencing gene regulation or metabolic activity in surrounding microbes or host cells. In the host context, methane has been described as a gasotransmitter that modulates intestinal motility and mechanistically contributes to slower gastrointestinal transit. The potential signaling functions of methane represent an intriguing and largely unexplored research direction, which should be addressed in future studies.

5. A previous study from Yachida et al reported that *M. smithii* was enriched in the late CRC; on the other hand, *F. nucleatum* was increased along with CRC progression. In this study, you validated the interaction of *M. smithii* and *F. nucleatum* by biological experiment; however, this interaction may not be observed in the cohort study. This is one of the limitations of this study.

We appreciate the reviewer's comment. Indeed, a previous multi-cohort analysis (<https://doi.org/10.1053/j.gastro.2024.10.023>) showed the increased abundance of *M. smithii* in early CRC as well (although not significant) and Yachida et al. (2019), reported that *Fusobacterium nucleatum* abundance increases with colorectal cancer (CRC) progression. To address this point, we have now included explicit stage-stratified analyses across the available CRC datasets. These reveal a consistent tendency for *M. smithii* to be elevated in advanced stages (III–IV), while some cohorts already show enrichment at earlier stages (e.g., stage 0 in Yachida) and others display moderate variability across stages I–II. Although limited by sample size, these results collectively support an overall increase of *M. smithii* abundance with tumor progression (now detailed in the revised manuscript, Lines 269 - 274).

We acknowledge that the strength of the *M. smithii*–*F. nucleatum* association may differ across cohorts due to differences in sequencing depth, cohort composition, and analytical coverage of archaeal reads. Nevertheless, added co-correlation analyses (Spearman's rho) show a consistent positive association between *M. smithii* and *F. nucleatum* in multiple datasets (e.g., Thomas et al., $p= 0.0103$), and the combined dataset (see new Supplementary Fig. 9). All this leads to the assumption that there might indeed be situations where *M. smithii* and *F. nucleatum* are co-localized, which is supported by our experimental co-culture validation, which demonstrates that *M. smithii* can modulate the growth and metabolic activity of *F. nucleatum* under controlled anaerobic conditions. Together, these findings suggest that while the interaction may not always be detectable at the community level in every dataset, it remains biologically plausible and mechanistically supported.

Nevertheless, our conclusions are therefore phrased conservatively: we do not claim that *M. smithii* drives CRC progression, but rather that its metabolic activity, through hydrogen consumption and syntrophic cross-feeding, can shape a microenvironment favorable for CRC-associated bacteria such as *F. nucleatum*, particularly in the later stages of disease when microbial dysbiosis and metabolic shifts become more pronounced.

Reviewer #3 (Remarks to the Author):

The authors have carried out a literature review and meta analysis of selected human gut microbiome studies to try to strengthen their claim that archaea, specifically *Methanobrevibacter smithii*, is a controlling contributor to a human disease. Of the thousands of studies, both large and small, that have been carried out on humans across the world from all age groups, they focus on twenty, two of which show a statistically significant correlation between disease and enriched archaeal species, however the effect on disease severity is not discussed.

We thank the reviewer for this important comment and the opportunity to clarify our dataset selection and analytical focus. We are fully aware that numerous human gut microbiome studies exist worldwide. However, as detailed in the manuscript and to avoid a “fishing exercise”, we deliberately focused on diseases previously linked to archaeal taxa, as summarized in our earlier reviews (<https://doi.org/10.1111/febs.17123>; <https://doi.org/10.1016/j.mib.2022.102146>), namely CRC, IBD, T2D, MS, AD, SCZ and PD (see Fig. 1 and references given in the introduction).

Further, to ensure accurate detection of archaeal taxa, we restricted our analysis to shotgun metagenomic datasets, thereby avoiding the known primer bias of 16S rRNA gene amplicon studies against Archaea (<https://doi.org/10.3389/fmicb.2019.02796>). This criterion, while stringent, improves confidence in archaeal detection and reduces the number of eligible datasets to those providing high-quality, quantitative profiles suitable for cross-cohort comparison. As outlined in the updated Figure 1, our selection prioritized large, well-annotated cohorts that enable robust meta-analysis rather than broad but heterogeneous inclusion.

The link of increased *M. smithii* signals with colorectal cancer has not only been observed by us, but by a number of other recent studies, including a very recent multi-cohort study that combined datasets using an alternative approach (DOI: 10.1053/j.gastro.2024.10.023). Without addressing the mechanistic question behind, several studies have reported this link, supporting our finding that *M. smithii* is indeed enriched in CRC.

In the revised version of the manuscript, we now provide additional analyses that include combined datasets for CRC, PD and pre-AD, all corrected for batch effects using MMUPHin (see updated Fig. 1 and related sections). Moreover, we have expanded our analyses to include disease severity where metadata permitted (CRC stages), revealing a consistent trend of increasing *M. smithii* abundance with advancing CRC stage.

With these additions, we hope we are able to convince you on our targeted strategy to understand the potential contribution of archaea to disease.

The correlation studies would be strengthened by including machine learning tools for multifactorial analysis to capture a more comprehensive picture of archaeal contribution to microbiome-associated digestive diseases, as is the standard in the field.

We thank the reviewer for this excellent suggestion. Indeed, several recent studies have already applied machine learning approaches to investigate microbiome- disease associations in particular in CRC, consistently identifying *Methanobrevibacter smithii* among the top discriminating taxa (e.g., <https://www.nature.com/articles/s41591-025-03693-9>). For this reason, we initially did not include a machine learning component, as our primary aim was to provide independent validation and mechanistic context for these findings using a combination of meta-analysis and experimental data.

Nevertheless, following the reviewer's helpful suggestion, we have now incorporated machine learning analyses (Random Forest classifier) in the revised manuscript to further strengthen the assessment of archaeal contributions, particularly to CRC. This approach allows evaluation of multifactorial feature importance, complementing our correlation-based findings. Consistent with previous reports, *M. smithii* again emerged among the top-ranked taxa contributing to CRC classification. This is now included in the manuscript.

We focused this new analysis on the CRC datasets, which provided the largest and most balanced sample size suitable for robust model training and evaluation. The details and results are now included in the revised manuscript (Lines 289-305 and Supplementary Fig. 7).

The authors then seek to bolster the claim that archaea promote colorectal cancer by culturing them with known cancer-promoting bacteria and characterizing metabolites in vitro. Unfortunately there are systematic methodological choices made by the investigators that affect the data interpretations. Rigorous culturing and phenomics experiments are needed to conclude how archaea interact with cancer-promoting bacteria, however the authors focus on one set of experiments that require enrichment of archaea to begin with – likely skewing the outcomes and thus overplaying the interpretations with respect to disease contribution.

We appreciate the reviewer's comments and fully understand the concern regarding potential biases introduced by the procedures for our co-culture experiments. Indeed, co-culturing bacteria with archaea remains technically challenging, and only a few studies to date have attempted such systems. As also practiced in other studies (e.g. Ruaud et al., co-culturing *Christensenella* with *Methanobrevibacter*), a (longer) pre-incubation of *Methanobrevibacter* was necessary to achieve comparable and sufficient starting biomass between partners. Different experimental set-ups were tested beforehand, and this procedure detailed in the manuscript assured the most reliable and reproducible outcomes.

We agree that such co-cultures are simplified and somewhat artificial systems, and we have been careful not to overinterpret their implication for disease. Our goal was not to claim a direct cancer-promoting role of archaea, but rather to explore mechanistic interactions that may influence bacterial metabolism relevant to CRC. To strengthen these insights, we complemented the co-culture setup with metabolomic profiling and electron microscopy, providing initial phenotypic evidence of archaea - bacterial metabolic interactions. We fully agree that further phenomics approaches, including e.g. transcriptomics will be essential for even deeper mechanistic understanding.

We now explicitly acknowledged in the revised discussion that our *in vitro* model represents a reduced approximation of the complex gut environment. Our revised text emphasizes that while the system applied simplifies natural conditions, it effectively is able to test archaeal-bacterial interactions under controlled anoxic settings.

Evidence of physical co-association between archaea and cancer-promoting bacteria are lacking, making it difficult to understand how secretion and consumption of pro- and anti-cancer metabolites by the microbes in a crowded gut environment would contribute to establishing disease. The manuscript would be strengthened by accounting for the mechanisms of cancer development caused by bacteria, similar to how other researchers have shown that archaea promote the impact of frank pathogens on health.

We thank the reviewer for this important comment and fully agree that direct evidence of physical co-association between *Methanobrevibacter smithii* and CRC-promoting bacteria remains limited. Demonstrating such spatial co-localization in human tissues represents a major technical challenge and remains missing even for many bacterial species strongly implicated in CRC. Establishing archaeal co-localization will therefore be an essential focus for future *in situ* studies.

Nevertheless, several lines of evidence from our work and the current literature support the biological plausibility of close archaeal - bacterial interactions in the CRC context. In the revised manuscript, we have expanded our analyses to include co-correlation and machine learning approaches, which together identify *M. smithii* as positively associated with multiple CRC-promoting bacterial taxa across cohorts (see new Supplementary Fig. 9, and Supplementary Fig. 10). Importantly, other recent large-scale multi-cohort meta-analyses (e.g. <https://doi.org/10.1053/j.gastro.2024.10.023>) also reported co-occurrence of *M. smithii* with CRC-associated bacteria, reinforcing the robustness of this observation. Furthermore, *M. smithii* and *Fusobacterium nucleatum* have both been detected in CRC tumor biopsies (<https://doi.org/10.1007/s00535-014-0963-x>), suggesting potential physical proximity within the tumor microenvironment.

Our interpretation is therefore not that *M. smithii* directly causes cancer, but that it may modulate the microbial ecosystem by facilitating the growth and metabolic activity of CRC-associated

bacteria through hydrogen consumption and syntrophic cross-feeding. This activity can enhance the production of bacterial metabolites linked to CRC pathogenesis, thus indirectly contributing to a tumor-promoting environment.

While we acknowledge the current lack of direct microscopic or spatial data, our expanded computational and experimental results, together with supporting evidence from recent literature, strongly suggest that *M. smithii* participates in a cooperative microbial network that favors CRC-associated bacterial functions. We have clarified this interpretation and its limitations in the revised Discussion section.

Reviewer #4 (Remarks to the Author):

I have been asked by Nature Comm to review the response to reviewers for this manuscript. I read with great interest the manuscript examining shotgun sequencing data from a number of studies pooled to examine the potential relationship between methanogens and colorectal cancer. The authors describe the association between methanogens and CRC cases in case-control studies and then go on to discuss the effects on the bacterial microbiome. Specifically, the authors find that methanogens enhance or appear to be associated with an increase in bacterial players that have known pathogenic implications in progression to CRC such as fusobacterium, E. coli, and others. Co-culture studies were helpful in showing that methanogens could encourage the growth of these other organisms in simplistic growth simulations. They further go on to study the supernatant for these experiments seeming to find some biochemical components that have published relationships to mucosal health and CRC development as well.

Overall, the authors received 3 reviews with two of the three reviews having minor commentary and corrections which the authors handled well. The final reviewer was more concerned about the overreaching arguments. Specifically, there was a concern about the use on only a few studies of the many published over time. The authors appear to sufficiently address the reasons for their selection of databases (primarily finding databases that use shotgun sequencing sufficient enough to assess for archaea.

More importantly, the reviewer appeared to be suggesting a more guarded tone to the association with CRC. The study is not a cause and effect study. It is an association study. Methanogens are more associated with CRC. The presence of methanogens appear to draw or encourage a greater diversity and consortium of bacteria some of whom have associations with CRC. They prove the encouragement of growth in co-culture and some of the byproducts of their co-culture were interesting molecules that could have importance based on the previous literature in CRC development. But all of this is associative and not cause and effect.

I think the paper is very important and warrants publication. I think the author have toned down any language around cause and effect. I do believe they could do a bit more to tone this down. This paper does not say that methanogens cause cancer and hopefully is not reading in that way. In fact, data suggest methanogens may be associated with healthy aging and maybe even longevity. But, it could be strain dependent for methanogens as well as epigenetic interrelationships that are yet to be determined. Thus being introspective until the next round of data can draw more direct lines would be important.

Authors' response: We would like to thank the reviewer for valuable feedback. We appreciate it. We have once more read the manuscript and toned town the language further. We have also addressed your insight in the discussion section.